# SCALING LAWS FOR DIFFUSION TRANSFORMERS

**Zhengyang Liang**[1,2]   **Hao He**[3]   **Ceyuan Yang**[3*]   **Bo Dai**[4*†]
[1]University of Toronto    [2]Vector Institute
[3]The Chinese University of Hong Kong    [4]The University of Hong Kong

## ABSTRACT

Diffusion transformers (DiT) have already achieved appealing synthesis and scaling properties in content recreation, *e.g.,* image and video generation. However, scaling laws of DiT are less explored, which usually offer precise predictions regarding optimal model size and data requirements given a specific compute budget. Therefore, experiments across a broad range of compute budgets, from `1e17` to `6e18` FLOPs are conducted to confirm the existence of scaling laws in DiT *for the first time*. Concretely, the loss of pretraining DiT also follows a power-law relationship with the involved compute. Based on the scaling law, we can not only determine the optimal model size and required data but also accurately predict the text-to-image generation loss given a model with 1B parameters and a compute budget of `1.5e21` FLOPs. Additionally, we also demonstrate that the trend of pretraining loss matches the generation performances (*e.g.,* FID), even across various datasets, which complements the mapping from compute to synthesis quality and thus provides a predictable benchmark that assesses model performance and data quality at a reduced cost.

## 1   INTRODUCTION

Scaling laws in large language models (LLMs) (Kaplan et al., 2020; Hestness et al., 2017; Henighan et al., 2020; Hoffmann et al., 2022) have been widely observed and validated, suggesting that pretraining performance follows a power-law relationship with the compute $C$. The actual compute could be roughly calculated as $C = 6ND$, where $N$ is the model size and $D$ is the data quantity. Therefore, determining the scaling law helps us make informed decisions about resource allocation to maximize computational efficiency, namely, figure out the optimal balance between model size and training data (*i.e.,* the optimal model and data scale) given a fixed compute budget. However, scaling laws in diffusion models remain less explored.

The scalability has already been demonstrated in diffusion models, especially for diffusion transformers (DiT). Specifically, several prior works (Mei et al., 2024; Li et al., 2024) reveal that larger models always result in better visual quality and improved text-image alignment. However, the scaling property of diffusion transformers is *clearly observed but not accurately predicted*. Besides, the absence of explicit scaling laws also hinders a comprehensive understanding of how training budget relate to model size, data quantity, and loss. As a result, we cannot determine accordingly the optimal model and data sizes for a given compute budget and accurately predict training loss. Instead, heuristic configuration searches of models and data are required, which are costly and challenging to ensure optimal balance.

In this work, we characterize the scaling behavior of diffusion models for text-to-image synthesis, resulting in the explicit scaling laws of DiT *for the first time*. To investigate the explicit relationship between pretraining loss and compute, a wide range of compute budgets from `1e17` to `6e18` FLOPs are used. Models ranging from 1M to 1B are pretrained under given compute budgets. As shown in Fig. 1(a), for each compute budget, we can fit a parabola and extract an optimal point that corresponds to the optimal model size and consumed data under that specific compute constraint. Using these optimal configurations, we derive scaling laws by fitting a power-law relationship between compute budgets, model size, consumed data, and training loss. To evaluate the derived

---

*Equal Supervision.
†Corresponding Author.

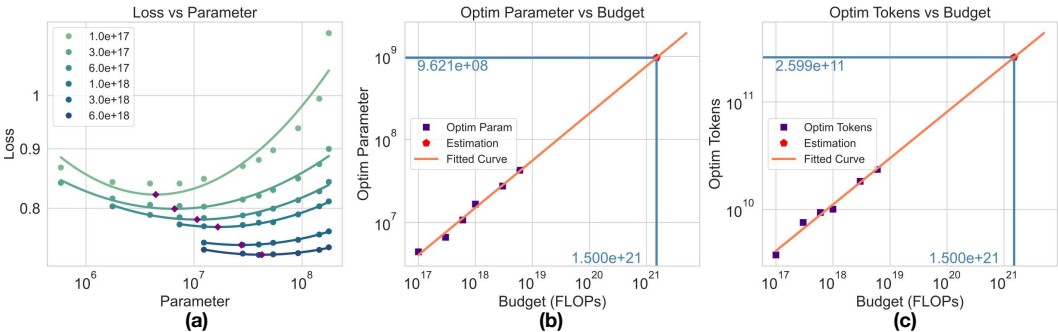

Figure 1: **IsoFLOP and Model/Data Scaling Curves.** For each training budget, we train multiple models of varying sizes. A parabola is fitted to the loss for each training budget, and the minimum point on each parabola (represented by the purple dots) corresponds to the optimal allocation of model size and data for that specific budget. By identifying the model and data sizes at these optimal points, we can plot the scaling trends of model parameters, tokens, and training budgets. The power-law curves shown allow us to predict the optimal configurations for larger compute budgets, such as `1.5e21` FLOPs.

scaling laws, we extrapolate the compute budget to `1.5e21` FLOPs that results in the compute-optimal model size (approximately 1B parameters) and the corresponding data size. Therefore, a 1B-parameter model is trained under this budget and the final loss matches our prediction, demonstrating the effectiveness and accuracy of our scaling laws.

To make the best use of the scaling laws, we demonstrate that the generation performances (*e.g.,* FID (Fréchet Inception Distance)) also match the trend of pretraining loss. Namely, the synthesis quality also follows the power-law relationship with the compute budget, making it predictable. More importantly, this observation is transferable across various datasets. We conduct additional experiments on the COCO validation set (Lin et al., 2014), and the same scaling patterns hold, even when tested on out-of-domain data. Accordingly, scaling laws could serve as a predictable benchmark where we can assess the quality of both models and datasets at a significantly reduced computational cost, enabling efficient evaluation and optimization of the model training process.

To summarize, We at first confirm the presence of scaling laws in diffusion transformers during training, revealing a clear power-law relationship between compute budget and training losses. Next, we establish a connection between pretraining loss and synthesis evaluation metrics. Finally, We conducted preliminary experiments that demonstrate the potential of using scaling laws to evaluate both data and model performance. By conducting scaling experiments at a relatively low cost, we can assess and validate the effectiveness of different configurations based on the fitted power-law coefficients.

## 2 RELATED WORK

**Scaling Laws**   Scaling laws (Hestness et al., 2017) have been fundamental in understanding the performance of neural networks as they scale in size and data. This concept has been validated across several large pretraining models (Dubey et al., 2024; Bi et al., 2024; Achiam et al., 2023). Kaplan et al. (2020); Henighan et al. (2020) were the first to formalize scaling laws in language models and extend them to autoregressive generative models, demonstrating that model performance scales predictably with increases in model size and dataset quantity. Hoffmann et al. (2022) further highlighted the importance of balancing model size and dataset size to achieve optimal performance. In the context of diffusion models, prior works (Mei et al., 2024; Li et al., 2024) have empirically demonstrated their scaling properties, showing that larger compute budgets generally result in better models. These studies also compared the scaling behavior of various model architectures and explored sampling efficiency. Hu et al. explains the properties of diffusion transformers from a statistical perspective, where the approximation and generalization theory respectively support the scalability of diffusion transformers in terms of model and data. However, no previous works provide an explicit formulation of scaling laws for diffusion transformers to capture the relationship

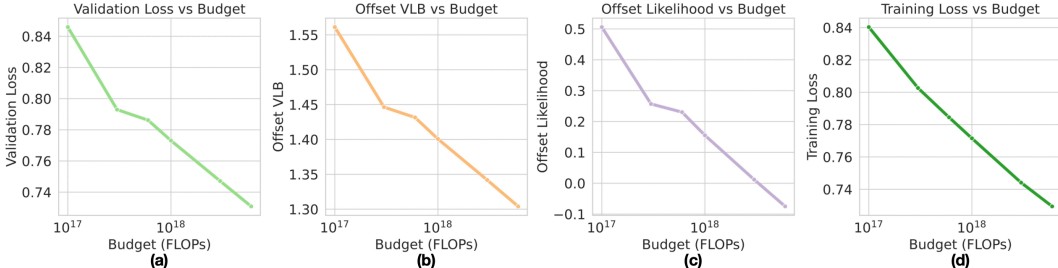

Figure 2: **Scaling Curves for Different Metrics.** We present the scaling curves for training and validation loss, offset VLB, and offset likelihood. Validation metrics are evaluated on the COCO 2014 validation set. The metrics display consistent trends and similar shapes, all adhering to a power-law relationship. This demonstrates that each of these metrics can be used to observe scaling laws effectively. For simplicity, we primarily focus on training loss in subsequent analyses.

between compute budget, model size, data, and loss. In this paper, we aim to address this gap by systematically investigating the scaling behavior of diffusion transformers (DiTs), offering a more comprehensive understanding of their scaling properties.

## 3 METHOD

### 3.1 BASIC SETTINGS

Unless otherwise stated, all experiments in this paper adhere to the same basic settings. While the training techniques and strategies employed may not be optimal, they primarily affect the scaling coefficients rather than the scaling trends. In this section, we outline the critical settings used in our experiments. Additional details are provided in Appendix E.

**Diffusion Formulation** All experiments are conducted using the Rectified Flow (RF) formulation (Liu et al., 2022; Lipman et al., 2022; Albergo et al., 2023) with $\mathbf{v}$-prediction. For timestep sampling, we adopt the Logit-Normal (LN) Sampling scheduler $\pi_{ln}(t; m, s)$, as proposed in Esser et al. (2024). A detailed ablation study of this choice can be found in Appendix H.1.

**Models & Dataset** As observed in Kaplan et al. (2020), model design has limited influence on scaling behavior unless the architecture is extremely shallow or narrow. We adopt a vanilla transformer architecture (Vaswani, 2017) with minimal modifications. Input tokens—comprising text, image, and timestep embeddings—are concatenated following in-context conditioning (Peebles & Xie, 2023). Our dataset consists of 108 million image-text pairs randomly sampled from Laion-Aesthetic (Schuhmann et al., 2022), and re-captioned using LLAVA 1.5 (Liu et al., 2024). A validation set of 1 million samples is drawn from the same subset. Further details are provided in Appendix E.1 and E.2. In most of our experiments, each data point is seen only once during training, which is consistent with the data-infinite setting commonly adopted in the industry. However, scaling laws do not rely on this assumption. To demonstrate this, we also evaluate scaling behavior under a data-constrained setting using ImageNet (Deng et al., 2009), with results presented in Appendix H.9.

### 3.2 SCALING METRICS

A natural question arises when investigating scaling laws during training: *What metrics should be selected to observe scaling behavior?* In the context of Large Language Models (LLMs), the standard approach is autoregressive training (Radford, 2018; Radford et al., 2019), where the model is trained to predict the next token in a sequence, directly optimizing the likelihood of the training data. This has proven to be a reliable method for measuring model performance as the compute budget scales up. Inspired by this approach, we extend the concept of scaling laws to diffusion models, using **loss** and **likelihood** as our key metrics.

### 3.2.1 Loss

Loss is the primary metric chosen to observe scaling behavior during training. Unlike autoregressive models, diffusion models do not directly optimize likelihood. Instead, the objective is to match a time-conditioned velocity field (Ma et al., 2024b; Liu et al., 2022; Lipman et al., 2022; Albergo et al., 2023). Specifically, the velocity $\mathbf{v}(x_t, t)$ at timestep $t$ is defined as:

$$\mathbf{v}(x_t, t) = x'_t = \alpha'_t x_0 + \beta'_t \epsilon, \tag{1}$$

where $x_0$ represents the original data and $\epsilon$ denotes the noise. Here, the prime symbol $'$ indicates the derivative with respect to time $t$. In the rectified flow framework, the coefficients $\alpha_t$ and $\beta_t$ are defined as $\alpha_t = 1 - t, \beta_t = t$. Thus, the velocity $\mathbf{v}$ can be further simplified as:

$$\mathbf{v}(x_t, t) = -x_0 + \epsilon. \tag{2}$$

The corresponding loss function is expressed in terms of the expected value:

$$\mathcal{L}(\theta) = \mathbb{E}_{\mathbf{x}_0 \sim p_{\text{data}}, \, t \sim \mathcal{U}(\{1,\ldots,T\}), \, \epsilon \sim \mathcal{N}(\mathbf{0}, \mathbf{I})} \left[ \left\| \mathbf{v}_\theta(\mathbf{x}_t, t) + \mathbf{x}_0 - \epsilon \right\|^2 \right] \tag{3}$$

$$\approx \frac{1}{N} \sum_{i=1}^{N} \left\| \mathbf{v}_\theta(\mathbf{x}_{t_i}^{(i)}, t_i) + \mathbf{x}_0^{(i)} - \epsilon_i \right\|^2, \tag{4}$$

The **training loss** is estimated using a Monte Carlo method, which involves timesteps and noise sampling. In Eq. 4, $N$ denotes the number of training samples in the mini-batch, $t_i$ and $\epsilon_i$ are the timestep and noise for constructing the i-th sample. The stochasticity inherent in this process can cause significant fluctuations, which are mitigated by employing a larger batch size of 1024 and applying Exponential Moving Average (EMA) smoothing on the loss value. In our experiments, we set $\alpha_{\text{EMA}} = 0.9$, which is found to produce stable results. A detailed ablation study on the choice of loss EMA coefficients is provided in Appendix H.2. This smoothing procedure helps reduce the variance of loss value and provides clearer insights into training dynamics.

In addition to the training loss, **validation loss** is also computed on the COCO 2014 dataset (Lin et al., 2014). To ensure consistency with the training loss, timesteps are sampled using the LN timestep sampler $\pi_{ln}(t; m, s)$, and evaluation is performed on 10,000 data points, with 1,000 timesteps sampled per point.

### 3.2.2 Likelihood

Likelihood is our secondary metric. The likelihood over the dataset distribution $P_\mathcal{D}$ given model parameters $\theta$ is represented as $\mathbb{E}_{x \sim P_\mathcal{D}}[p_\theta(x)]$, which can be challenging to compute directly. In this paper, we measure likelihood using two different methods. The first method is based on the Variational Autoencoder (VAE) framework (Kingma et al., 2021; Song et al., 2021; Vahdat et al., 2021), which approximates the lower bound of log-likelihood using the Variational Lower Bound (VLB). Since the VAE component in our experiments is fixed to Stable Diffusion 1.5, terms related to the VAE remain constant and are ignored in our computation, as they do not affect the scaling behavior. The second method uses Neural Ordinary Differential Equations (ODEs) (Chen et al., 2018; Grathwohl et al., 2018), enabling the computation of exact likelihood through reverse-time sampling.

**Variational Lower Bound (VLB)** We estimate the VLB following the approach in Kingma et al. (2021), where it is treated as a reweighted version of the validation loss. The process starts by converting the estimated velocity into a corresponding estimate of $x_0$, after which the loss is computed based on the difference between the estimated $x_0$ and the original sample. To obtain the VLB, this loss is further reweighted with the weighting coefficient being the derivative of signal-to-noise ratio of noisy samples with respect to time $t$, *i.e.,* $\text{SNR}'(t)$. More details can be found in Appendix F. All models are evaluated on the COCO 2014 validation set using 10,000 data pairs. For each data point, 1,000 timesteps are sampled to ensure accurate estimations of the VLB.

**Exact Likelihood** The exact likelihood is computed using reverse-time sampling, where a clean sample is transformed into Gaussian noise. The accumulated likelihood transition is calculated using

the instantaneous change of variables theorem (Chen et al., 2018):

$$\log p_\theta(x) = \log p_\theta(x_T) - \int_0^T \nabla \cdot f_\theta(x_t, t)\, dt, \tag{5}$$

where $f_\theta(x_t, t)$ represents the model's output, and $\log p_\theta(x_T)$ is the log density of the final Gaussian noise. The reverse process evolves $t$ from 0 (data points) to $T$ (noise). The reverse sampling is performed over 500 steps using the Euler method.

Following model training, we evaluate models on the validation set to assess their compute-optimal performance. As illustrated in Fig. 2, all four metrics (training loss, validation loss, offset VLB, and offset exact likelihood ) exhibit similar trends and shapes as the training budget increases, showing their utility in observing scaling laws. All metrics for all models are computed using the evaluation protocol described in Sec. 3.3. These consistent patterns suggest that any of these metrics can be effectively used to monitor scaling behavior. However, to simplify our experimental workflow, we prioritize **training loss** as the primary metric. Training loss can be observed directly during the training process, without the need for additional evaluation steps, making it a more practical choice for tracking scaling laws in real-time.

### 3.3 SCALING LAWS IN TRAINING DIFFUSION TRANSFORMERS

**Scaling Laws**  In this section, we investigate the scaling laws governing diffusion transformers, which describe the relationships between several key quantities: the objective function, model parameters, tokens, and compute budget. The objective measures discrepancy between the data and model's predictions. The number of parameters $N$ reflects model's capacity, while tokens $D$ denote the total amount of data (in tokens) processed during training. Compute budget $C$ (we also refer to C as compute), typically measured in Floating Point Operations (FLOPs), quantifies the total computational resources consumed. In our experiments, the relationship between compute, tokens, and model size is formalized as $C = 6ND$, directly linking the number of parameters and tokens to the overall compute budget. We estimate $C$ by explicitly counting the floating-point operations required by the transformer blocks of our diffusion transformer. Focusing on these blocks, which dominate the total cost, and accounting for both forward and backward passes, we find that, on average, processing a single token requires approximately $6N$ FLOPs. Hence, training a model with $N$ parameters on $D$ tokens consumes

$$C = 6ND,$$

and in our setting, the compute budget, number of parameters, and tokens approximately satisfy this relationship. More details about FLOPs computation can be found in Appendix G and Sec. 2.1 in Kaplan et al. (2020).

For each compute budget $C$, we extract the corresponding compute-optimal parameter count $N_{\mathrm{opt}}(C)$ and token count $D_{\mathrm{opt}}(C)$ from configuration at the minimum point (purple marker) of each isoFLOP parabola in Fig. 1(a). When we plot these optimal points on log–log axes, both $\log N_{\mathrm{opt}}$ and $\log D_{\mathrm{opt}}$ vary approximately linearly with $\log C$, indicating an underlying power-law relationship between compute and the optimal allocation of parameters and data.Building on this, we hypothesize that power law equations can effectively capture the scaling relationships between these quantities. Specifically, we represent the optimal model size and token count as functions of the compute budget as follows:

$$N_{\mathrm{opt}} \propto C^a \quad \text{and} \quad D_{\mathrm{opt}} \propto C^b, \tag{6}$$

where $N_{\mathrm{opt}}$ and $D_{\mathrm{opt}}$ denote the optimal number of parameters and tokens for a given compute budget $C$, with $a$ and $b$ as scaling exponents that describe how these quantities grow with compute.

To empirically verify these scaling relationships, following Approach 2 in Hoffmann et al. (2022), we plot the isoFLOP figure to explore scaling laws. We select compute budgets [`1e17`, `3e17`, `6e17`, `1e18`, `3e18`, `6e18`]. We change the In-context Transformers from 2 layers to 15 layers. For all experiments, we use AdamW (Kingma, 2014; Loshchilov, 2017) as the default optimizer with a constant learning rate of 1e-4. Following several common practice from large-scale models Wortsman et al. (2023); Team (2024); Molybog et al. (2023), we use a weight decay of 0.01, an epsilon value of `1e-15`, and betas [0.9, 0.95]. For all experiments, we employ a batch size of

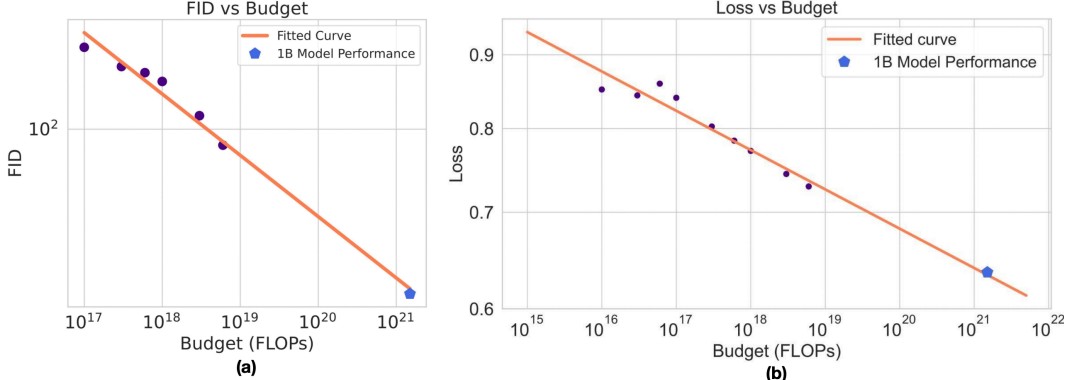

Figure 3: **Performance Scaling Curves.** The plot shows the relationship between training loss, FID, and compute budget, with both axes displayed on a logarithmic scale. The orange lines represent the fitted power-law curves for both metrics across various budgets, while the purple dots mark the performance of compute-optimal models at smaller budgets. In both cases, the blue pentagons indicate the predicted performance at a budget of `1.5e21` FLOPs, demonstrating highly accurate predictions for large-scale models.

1024 and apply gradient clipping with a threshold of 1.0. To enable classifier-free guidance (Ho & Salimans, 2022), we randomly drop the label with a probability of 0.1. During training, we use mixed precision with the BF16 data format. As is standard in diffusion models, we also maintain an EMA version of the model with a decay coefficient of 0.99, which is saved for evaluation. The scaling curves of the EMA model are presented in Appendix H.4.

For each budget, we fit a parabola to the resulting performance curve, as illustrated in the Fig. 1(a), to identify the optimal loss, model size, and data allocation (highlighted by the purple dots). By collecting the optimal values from different budgets and fitting them to a power law, we establish relationships between the optimal loss, model size, data, and compute budgets.

As shown in the Fig. 1(a), except for the `1e17` budget, the parabolic fits align closely with the empirical results. This analysis confirms that the optimal number of parameters and tokens scale with the compute budget according to the following expressions:

$$N_{\text{opt}} = 0.0009 \cdot C^{0.5681}, \tag{7}$$

$$D_{\text{opt}} = 186.8535 \cdot C^{0.4319}. \tag{8}$$

In this way, we establish the relationship between compute budget and model/data size. And from the fitted scaling curves, we observe that the ratio between the scaling exponent for data and the scaling exponent for model size is $0.4319/0.5681$. This indicates that, under our specific settings, both the model and data sizes need to increase in tandem as the training budget increases. However, the model size grows at a slightly faster rate compared to the data size, as reflected by the proportional relationship between the two exponents.

Additionally, in Fig. 3(b), we fit the relationship between the compute budget and loss, which follows the equation:

$$L = 2.3943 \cdot C^{-0.0273}. \tag{9}$$

To validate the accuracy of these fitted curves, we calculate the optimal model size (958.3M parameters) and token count for a compute budget of `1.5e21`. A model is then trained with these specifications to compare its training loss with the predicted value. As demonstrated in Fig. 3(b), this model's training loss closely matches the predicted loss, further confirming the validity of our scaling laws.

### 3.4 PREDICTING GENERATION PERFORMANCE

Evaluating generative models has long been a challenge, as many commonly used metrics have been criticized for their limitations. However, recent work (Esser et al., 2024; Fan et al., 2025) on large-

scale foundation generative models has shown that metrics such as FID, GenEval, and loss, despite their imperfections, correlate well with human perception and preference—widely considered the gold standard in image generation evaluation. Notably, Esser et al. (2024) reports that the diffusion transformer's loss serves as a strong predictor of overall model quality. In previous sections, we have analyzed the scaling behavior of loss. Here, we extend our analysis to additional evaluation metrics. Specifically, we present scaling curves for **FID**, **GenEval**, and **human preferences**. Due to space constraints, the GenEval and Human preferences results are included in Appendix H.10 and H.11. These results confirm that generation quality also scales predictably with compute, following a power-law trend. We also provide some visual samples generated by compute-optimal models in Appendix H.8.

**Evaluation Metrics**    We evaluate generation quality using multiple complementary metrics: FID, GenEval, and human preference reward models (HPSv2.1 (Wu et al., 2023) and ImageReward (Xu et al., 2023)). FID quantifies the distance between the distributions of generated and real images in a feature space, with lower values indicating higher fidelity. Following Sauer et al. (2021), we compute FID using CLIP features (ViT-L/14 Dosovitskiy et al. (2020)) instead of the traditional Inception features. GenEval captures the capability of instance-level compositionality by employing object detectors to provide fine-grained analyses, such as color binding and object counting. Finally, human preference reward models (HPSv2.1 and ImageReward) approximate human judgments by scoring generations according to preference signals.

**Scaling laws for FID Predictions**    In this section, we provide analysis for FID. We reveal that the relationship between FID and the training compute budget follows a clear power-law trend, as shown in Fig. 3(b). The relationship is captured by the following equation:

$$\texttt{FID} = 2.2566 \times 10^6 \cdot C^{-0.234}, \tag{10}$$

where $C$ is the training compute budget. The purple dots in the figure represent the FID scores of compute-optimal models at various budgets, and the orange line represents the fitted power-law curve. Notably, the prediction for FID at a large budget of `1.5e21` FLOPs (blue pentagon) is highly accurate, confirming the reliability of scaling laws in forecasting model performance even at larger scales. As the compute budget increases, FID values decrease consistently, demonstrating that scaling laws can effectively model and predict the quality of generated images as resources grow.

### 3.5    SCALING LAWS FOR OUT-OF-DOMAIN DATA

Scaling laws remain valid even when models are tested on out-of-domain datasets. To demonstrate this, we conduct validation experiments on the COCO 2014 validation set (Lin et al., 2014), using models that were trained on the Laion5B subset. In these experiments, we evaluate four key metrics: validation loss, Variational Lower Bound (VLB), exact likelihood, and Fréchet Inception Distance (FID). Each metric is tested on 10,000 data points to examine the transferability of the scaling laws across datasets. Additional experiments on other datasets (Flickr30k (Plummer et al., 2016), JourneyDB (Sun et al., 2024)) are presented in Appendix H.7.

The results, as shown in Fig. 4, reveal two key observations:

- **Consistent Trends**: Across all four metrics (validation loss, VLB, exact likelihood, and FID), the trends are consistent between the Laion5B subset and the COCO validation dataset. As the training budget increases, model performance improves in both cases, indicating that scaling laws generalize effectively across datasets, regardless of domain differences.

- **Vertical Offset**: There is a clear vertical offset between the metrics for the Laion5B subset and the COCO validation dataset, with consistently higher metric values observed on the COCO dataset. This suggests that while scaling laws hold, the absolute performance is influenced by dataset-specific characteristics, such as complexity or distribution. For metrics like validation loss, VLB, and exact likelihood, this offset remains relatively constant across different training budgets. The gap between the FID values for the Laion5B subset and the COCO validation set widens as the training budget increases. However, the relationship between FID and budget on the COCO validation set still follows a power-law trend. This suggests that, even on out-of-domain data, the FID-budget relationship can be

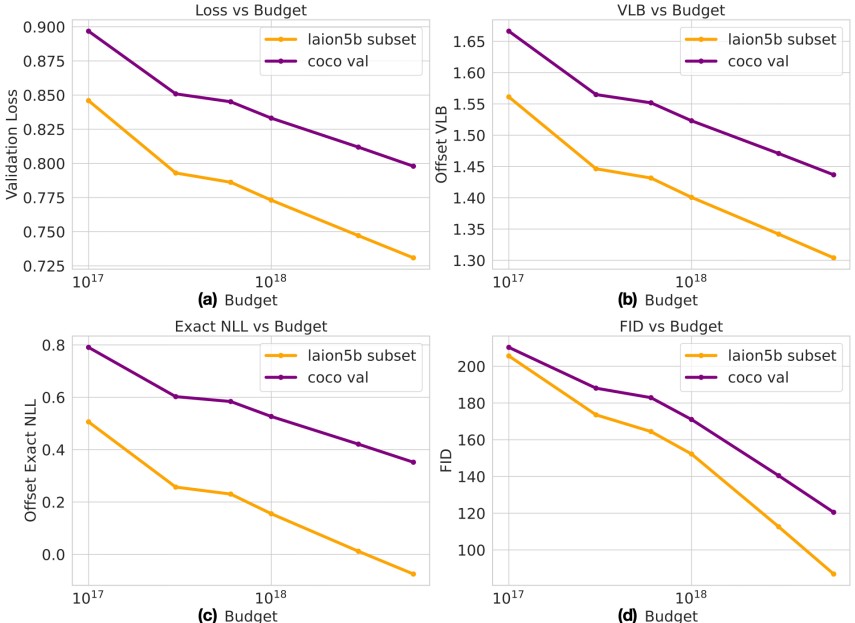

Figure 4: **Scaling laws for OOD data**. The evaluation of loss, VLB, and NLL are conducted on an out-of-domain dataset COCO 2014 validation set. All metrics decrease monotonically as the budget increases and they share a similar shape. However, a notable shift can be observed. Models have worse performance on the COCO validation set since they are trained on a different data distribution. The shift reflects the differences between datasets.

> reliably modeled using a power law, allowing us to predict the model's FID for a given budget.

In summary, these results demonstrate that scaling laws are robust and can be applied effectively to out-of-domain datasets, maintaining consistent trends while accounting for dataset-specific performance differences. Despite the vertical offset in absolute performance, particularly in metrics like FID, the power-law relationships remain intact, allowing for reliable predictions of model performance under varying budgets. These findings highlight the potential of scaling laws as a versatile tool for understanding model behavior across datasets. The ability to project model efficiency and performance onto new data domains underscores the broader applicability of scaling laws in real-world scenarios.

## 4 SCALING LAWS AS A PREDICTABLE SCALABILITY BENCHMARK

Scaling laws offer a robust framework for evaluating the design quality of both models and datasets, particularly with respect to their **scalability**. Previous work such as Dubey et al. (2024); Bi et al. (2024) have all explored the scaling laws in data mix and quality. By modifying either the model architecture or the data pipeline, isoFLOP curves can be generated at smaller compute budgets to assess the impact of these changes. Specifically, after making adjustments to the model or dataset, experiments can be conducted across a range of smaller compute budgets, and the relationship between compute and metrics such as loss, parameter count, or token count can be fitted. The effectiveness of these modifications can then be evaluated by analyzing the exponents derived from the power-law fits. Scaling laws provide a predictive framework for estimating a model's performance when scaled in both data and model size, offering a robust benchmark for evaluating the scalability of data and model designs.

Our evaluation follows three key principles:

- **Model improvements**: With a fixed dataset, a more efficient model will exhibit a lower model scaling exponent and a higher data scaling exponent. This suggests that the model

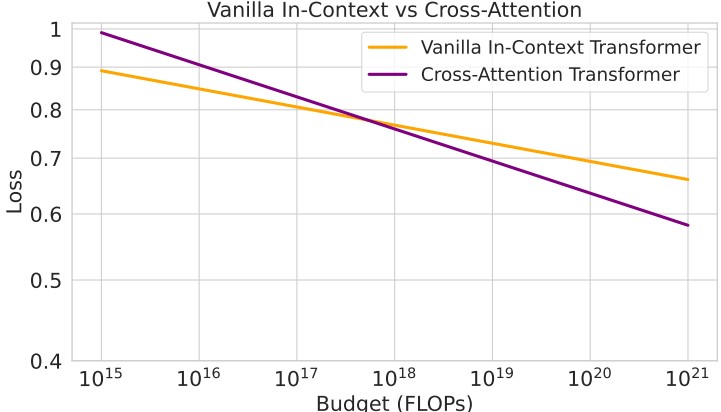

Figure 5: **Scaling curves for In-Context and Cross-Attention Transformers.** The plot compares the scaling behavior of In-Context Transformers and Cross-Attention Transformers with respect to compute budget (FLOPs). In-context transformers, which concatenate image, text, and time tokens, show a more gradual decline in loss compared to Cross-Attention Transformers, which inject time and text tokens via AdaLN and cross-attention blocks. The steeper slope of Cross-Attention Transformers indicates more efficient performance improvement with increasing compute, meaning they achieve lower loss with the same budget. These results highlight the efficiency of Cross-Attention Transformers and illustrate the potential of scaling laws in predicting performance trends across different model architectures.

can more effectively utilize the available data, allowing for a greater focus on increasing the dataset size with additional compute resources.

- **Data improvements**: When the model is fixed, a higher-quality dataset will result in a lower data scaling exponent and a higher model scaling exponent. This implies that a better dataset enables the model to scale more efficiently, yielding superior results with fewer resources.

- **Loss/FID improvements**: Across both model and data modifications, an improved training pipeline is reflected in a smaller loss/FID scaling exponent relative to compute. This indicates that the model achieves better performance with less compute, signaling overall gains in training efficiency.

To illustrate the utility of scaling laws as a predictive benchmark for scalability, we compare two specific transformer designs: (1) Vanilla In-Context Transformers, which concatenate image and visual tokens as input and utilize only standard self-attention blocks, and (2) Cross-Attention Transformers, where conditioning is incorporated through cross-attention mechanisms. We train and evaluate both models with a broad range of compute budgets, spanning 1e17, 3e17, 6e17, 1e18, 3e18, 6e18, and 1e19.

The scaling trends for both models are clearly illustrated in Fig. 5. The loss scaling curves demonstrate that as the compute budget increases, the performance of both models improves, but at different rates. The Cross-Attention Transformer shows a steeper decline in loss compared to the Vanilla In-Context Transformer, indicating that when scaled up, it achieves better performance with the same amount of compute.

The fitted scaling curves, as summarized in Tab. 1, support this observation. The Cross-Attention Transformer exhibits a smaller model exponent, meaning that as compute budgets increase, more resources should be allocated toward scaling the dataset. Additionally, the smaller loss exponent of the Cross-Attention Transformer suggests a more rapid decline in loss, indicating that this model achieves superior performance compared to the Vanilla In-Context Transformer. These findings align with the conclusions of Peebles & Xie (2023). However, this observation should not be generalized to conclude that the In-Context mechanism is inherently superior to Cross-Attention. Recent works, such as Flux and the MMDiT architecture in SD3 Esser et al. (2024), have demonstrated that

models utilizing In-Context Conditioning can outperform those relying on Cross-Attention. Our analysis does not aim to compare these mechanisms. Instead, the scaling laws benchmark serves as a tool for assessing model scalability within a given architecture. The method evaluates how efficiently a model benefits from increased data and computational resources as it scales. For completeness, we also provide a data quality comparison in Appendix H.6.

Table 1: Exponents of model, data, and loss for different model architectures.

| Model | Model Exponent | Data Exponent | Loss Exponent |
|---|---|---|---|
| Vanilla In-context | 0.56 | 0.43 | -0.0273 |
| Cross-Attention | 0.54 | 0.46 | -0.0385 |

This example illustrates how scaling laws can serve as a reliable and predictable scalability benchmark for evaluating the effectiveness of both model architectures and datasets. By analyzing the scaling exponents, we can draw meaningful conclusions about the potential and efficiency of different design choices in model and data pipelines.

## 5 CONCLUSION

In this work, we explored the scaling laws of Diffusion Transformers (DiT) across a broad range of compute budgets, from `1e17` to `6e18` FLOPs, and confirmed the existence of a power-law relationship between pretraining loss and compute. This relationship enables precise predictions of optimal model size, data requirements, and model performance, even for large-scale budgets such as `1e21` FLOPs. Furthermore, we demonstrated the robustness of these scaling laws across different datasets, illustrating their generalizability beyond specific data distributions. In terms of generation performance, we showed that training budgets can be used to predict the visual quality of generated images, as measured by metrics like FID. Additionally, by testing both In-context Transformers and Cross-Attention Transformers, we validated the potential of scaling laws to serve as a predictable benchmark for evaluating and optimizing both model and data design, providing valuable guidance for future developments in text-to-image generation using DiT.

## 6 ACKNOWLEDGMENTS

We would love to thank Tao Lu, Tianyi Lu for constructive feedback and help during the project. The project is supported in part by HKU Startup Fund and Institute of Data Science.

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

# A APPENDIX

## APPENDIX

This appendix is organized as follows:

- In Section B, we list an overview of the notation used in the paper.

- In Section D, we provide extended related work.

- In Section E, we provide experimental details.

- In Section F, we provide the derivation of the likelihood used in this paper.

- In Section G, we provide the details of scaling FLOPs counting based on our models.

- In Section H, we provide extra experiments.

- In Section I, we list the limitations and future works.

- In Section J, we discuss the in-context condition and cross attention mechanism in detail.

# B  NOTATION

Tab. B provides an overview of the notation used in this paper.

Table 2: **Summary of the notation and abbreviations used in this paper.**

| Symbol | Meaning |
|---|---|
| RF | Rectified Flow (Lipman et al., 2022; Liu et al., 2022; Albergo et al., 2023) |
| LN | Logit-Normal timestep sampler $\pi_{ln}(t; m, s)$ (Esser et al., 2024) |
| SNR | Signal-Noise-Ratio, $\lambda_t = \frac{\alpha_t^2}{\beta_t^2}$ |
| DDPM | Denoising Diffusion Probabilistic Models (Ho et al., 2020) |
| LDM | Latent Diffusion Models (Rombach et al., 2022) |
| VP | Variance Preserving formulation (Song et al., 2020) |
| VLB | Variational Lower Bound |
| VAE | Variational Autoencoder |
| KL | KL Divergence $KL(p(x)|q(x)) = \mathbb{E}_{p(x)}[ln\frac{p(x)}{q(x)}]$ |
| $P_{\mathcal{D}}$ | Dataset distribution |
| $d_{attn}$ | Dimension of the attention output |
| $d_{ff}$ | Dimension of the Feedforward layer |
| $d_{model}$ | Dimension of the residual stream |
| $n_{layer}$ | Depth of the Transformer |
| $l_{ctx}$ | Context length of input tokens |
| $l_{img}$ | Context length of image tokens |
| $l_{text}$ | Context length of text tokens |
| $l_{time}$ | Context length of time tokens |
| $N_{voc}$ | Size of Vocabulary list |
| $n_{head}$ | Number of heads in Multi-Head Attention |
| $N$ | Number of parameters |
| $D$ | Size of training data (token number) |
| $C$ | Compute budget, $C = 6ND$ |
| $N_{opt}$ | Optimal number of parameters for the given budget |
| $D_{opt}$ | Optimal training tokens for the given budget. |
| $\boldsymbol{\epsilon}$ | Gaussian Noise $\mathcal{N}(0, I)$ |
| $\mathbf{x}_t$ | A sample created at timestep $t$ |
| t | Timestep ranging from $[0, 1]$ |
| $\mathbf{v}$ | Velocity $\mathbf{v} = \mathbf{x}_1 - \mathbf{x}_0$ |
| $\alpha_t$ | $\alpha_t$ represents the scaling factor during noise sample creation. |
| $\beta_t$ | $\beta_t$ represents the diffusion factor during noise sample creation. |
| $\sigma_t$ | Noise Level defined for each timestep |
| $f_{\boldsymbol{\theta}}(\mathbf{x})$ | The network we use to learn the transition kernel. $f_{\boldsymbol{\theta}} : \mathbb{R}^{N \times M} \to \mathbb{R}^{N \times M}$ |
| $\eta$ | Learning rate |
| $\pi_{ln}(t; m, s)$ | Logit-Normal Timestep sampling schedule, $m$ is the location parameter, $s$ is the scale parameter |
| $\mathcal{L}(\boldsymbol{\theta}, \mathbf{x}, t)$ | Loss given model parameters, data points, and timesteps. |
| $\lambda$ | Fixed step size for ODE/SDE samplers |
| $\boldsymbol{\theta}$ | Parameters for diffusion models |
| $\boldsymbol{\phi}$ | Parameters for VAE encoder |
| $\boldsymbol{\psi}$ | Parameters for VAE decoder |
| $\alpha_{\text{EMA}}$ | EMA coefficient |

## C LLM USAGE

In this work, large language models (LLMs) were only employed to polish the wording and improve the readability of the manuscript. No part of the research design, data analysis, or result interpretation involved the use of LLMs. All scientific content and conclusions were produced entirely by the authors.

## D EXTENDED RELATED WORK

**Diffusion Transformers** Transformers have become the de facto model architecture in language modeling (Radford, 2018; Radford et al., 2019; Devlin, 2018), and they have also achieved significant success in computer vision tasks (Dosovitskiy et al., 2020; He et al., 2022). Recently, Transformers have been introduced into diffusion models (Peebles & Xie, 2023), where images are divided into patches (tokens), and the diffusion process is learned on these tokens. Additional conditions, such as timestep and text, are incorporated into the network via cross-attention (Chen et al., 2023), Adaptive Normalization (Perez et al., 2018), or by concatenating them with image tokens (Bao et al., 2023).Zheng et al. (2023) proposed masked transformers to reduce training costs, while Lu et al. (2024) introduced techniques for unrestricted resolution generation. Diffusion Transformers (DiTs) inherit the scalability, efficiency, and high capacity of Transformer architectures, positioning them as a promising backbone for diffusion models. Motivated by this scalability, we investigate the scaling laws governing these models in this work. To ensure robust and clear conclusions, we adopt a vanilla Transformer design (Vaswani, 2017), using a concatenation of image, text, and time tokens as input to the models.

**Diffusion Models** Diffusion models have gained significant attention due to their effectiveness in generative modeling, starting from discrete-time models (Sohl-Dickstein et al., 2015; Ho et al., 2020; Song & Ermon, 2019) to more recent continuous-time extensions (Song et al., 2020). The core idea of diffusion models is to learn a sequence of noise-adding and denoising steps. In the forward process, noise is gradually added to the data, pushing it toward a Gaussian distribution, and in the reverse process, the model learns to iteratively denoise, recovering samples from the noise. Continuous-time variants (Song et al., 2020) further generalize this framework using stochastic differential equations (SDEs), allowing for smoother control over the diffusion process. These methods leverage neural network architectures to model the score function and offer flexibility and better convergence properties compared to discrete versions. Diffusion models have shown remarkable success in various applications. For instance, ADM (Dhariwal & Nichol, 2021) outperforms GAN on ImageNet. Following this success, diffusion models have been extended to more complex tasks such as text-to-image generation. Notably, models like Stable Diffusion (Rombach et al., 2022) and DALLE (Ramesh et al., 2021) have demonstrated the ability to generate highly realistic and creative images from textual descriptions, representing a significant leap in the capabilities of generative models across various domains.

**Normalizing Flows** Normalizing flows has been a popular generative modeling approach due to their ability to compute exact likelihoods while providing flexible and invertible transformations. Early works like GLOW (Kingma & Dhariwal, 2018) and RealNVP (Dinh et al., 2016) introduced powerful architectures that allowed for efficient sampling and likelihood estimation. However, these models were limited by the necessity of designing specific bijective transformations, which constrained their expressiveness. To address these limitations, Neural ODE (Chen et al., 2018) and FFJORD (Grathwohl et al., 2018) extended normalizing flows to the continuous domain using differential equations. These continuous normalizing flows (CNFs) allowed for more flexible transformations by parameterizing them through neural networks and solving ODEs. By modeling the evolution of the probability density continuously, these methods achieved a higher level of expressiveness and adaptability compared to their discrete counterparts. Recent work has begun to bridge the gap between continuous normalizing flows and diffusion models. For instance, ScoreSDE (Song et al., 2020) demonstrated how the connection between diffusion models and neural ODEs can be leveraged, allowing both exact likelihood computation and flexible generative processes. More recent models like Flow Matching (Lipman et al., 2022) and Rectified Flow (Liu et al., 2022) further combined the strengths of diffusion and flow-based models, enabling efficient training via diffusion processes while maintaining the ability to compute exact likelihoods for generated samples. In this

paper, we build upon the formulation introduced by rectified flow and Flow Matching. By leveraging the training approach of diffusion models, we benefit from their generative performance, while retaining the capability to compute likelihoods.

**Likelihood Estimation**   Likelihood estimation in diffusion models can be approached from two primary perspectives: treating diffusion models as variational autoencoders (VAEs) or as normalizing flows. From the VAE perspective, diffusion models can be interpreted as models where we aim to optimize a variational lower bound (VLB) on the data likelihood (Kingma, 2013). The variational lower bound decomposes the data likelihood into a reconstruction term and a regularization term, where the latter measures the divergence between the approximate posterior and the prior. In diffusion models, this framework allows us to approximate the true posterior using a series of gradually noised latent variables. Recent works (Ho et al., 2020; Kingma et al., 2021; Song et al., 2021) have derived tighter bounds for diffusion models, enabling more accurate likelihood estimation by optimizing this variational objective. Alternatively, diffusion models can be viewed as a form of normalizing flows, particularly in the context of continuous-time formulations. Using neural ODEs (Chen et al., 2018), diffusion models can be trained to learn exact likelihoods by modeling the continuous reverse process as an ODE. By solving this reverse-time differential equation, one can directly compute the change in log-likelihood through the flow of probability densities (Grathwohl et al., 2018). This approach provides a method for exact likelihood computation, bridging the gap between diffusion models and normalizing flows, and offering a more precise estimate of the likelihood for generative modeling.

## E   EXPERIMENTAL DETAILS

### E.1   DATA

We primarily utilized three datasets in our work. Several ablation studies on formulation and model design were conducted using JourneyDB (Sun et al., 2024). Additionally, we curated a subset of 108 million image-text pairs from the Laion-Aesthetic dataset, applying a threshold of 5 for the aesthetic score. After collecting the data, we re-captioned all images using LLAVA 1.5 (Liu et al., 2024), specifically employing the LLaVA-Lightning-MPT-7B-preview model for this task. We then split the data into training and validation sets with a ratio of 100:1. Our third dataset is COCO (Lin et al., 2014), where we used the 2014 validation set to test scaling laws on an out-of-domain dataset.

### E.2   MODEL DESIGN

In this paper, we evaluate two distinct model architectures. For the PixArt model, we follow the original design presented in Chen et al. (2023). The In-Context Transformers are based on the In-Context block described in Peebles & Xie (2023). To facilitate large-scale model training, we employ QK-Norm (Dehghani et al., 2023) and RMSNorm (Zhang & Sennrich, 2019). The patch size is set to 2. Although previous work (Kaplan et al., 2020) suggests that the aspect ratio (width/depth) of Transformers does not significantly impact scaling laws, it is crucial to maintain a consistent ratio when fitting models to scaling laws. To demonstrate this, we train a series of models under a fixed computational budget, selecting models of various sizes and aspect ratios (32 and 64). We then plot the relationship between the number of parameters and loss. As illustrated in Fig. 6, mixing models with aspect ratios of 64 and 32 obscure the overall trend. To address this issue, we maintain the aspect ratio at 64 throughout.

## F   DERIVATION OF THE LIKELIHOOD

In this section, we provide a derivation of the likelihood estimation in our paper. In this paper, we use two ways to compute the likelihood. The first method is estimating the VLB (variational lower bound). Following Kingma et al. (2021); Vahdat et al. (2021), we derive a VLB in latent space. However, we cannot compute the entropy terms in the VAE. So our surrogate metric differs from the true VLB up to a constant factor.

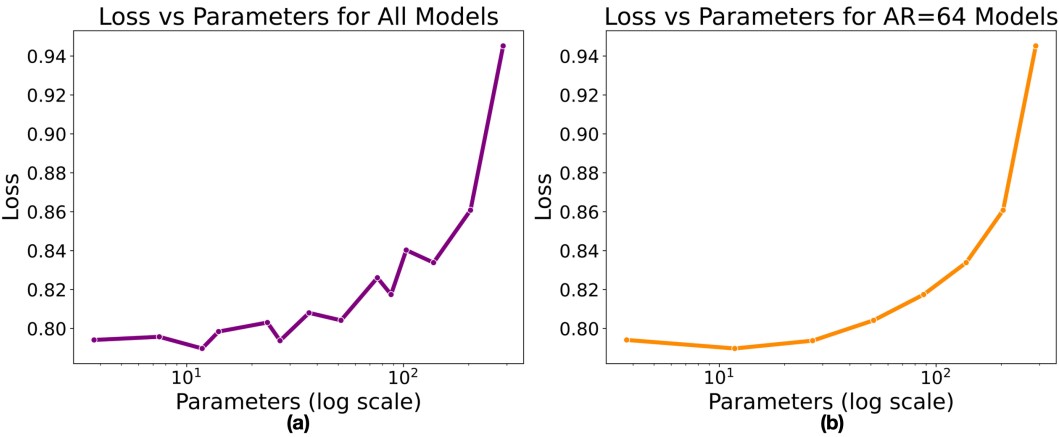

Figure 6: The effect of AR.

**VAE** The latent diffusion model (Vahdat et al., 2021; Rombach et al., 2022) consists of two components: a variational autoencoder (VAE) that encodes images into a latent space and decodes latents back into images, and a continuous diffusion model that operates in the latent space. To train the latent diffusion model, we optimize the variational encoder $q_\phi$, the decoder $p_\psi$, and the diffusion model $p_\theta$. Following Ho et al. (2020), the models are trained by minimizing the variational upper bound on the negative log-likelihood $\log P(x)$:

$$\mathcal{L}_{\theta,\phi,\psi}(x) = \mathbb{E}_{q_\phi(z_0|x)}[-\log p_\psi(x|z_0)] + KL(q_\phi(z_0|x)||p_\theta(z_0)) \tag{11}$$

$$= \mathbb{E}_{q_\phi(z_0|x)}[-\log p_\psi(x|z_0)] + \mathbb{E}_{q_\phi(z_0|x)}[\log q_\phi(z_0|x)] + \mathbb{E}_{q_\phi(z_0|x)}[-\log p_\theta(z_0)]. \tag{12}$$

In our implementation, we directly adopt the VAE from Stable Diffusion 1.5 and keep it fixed during training. As a result, the reconstruction term (first term) and the negative encoder entropy term (second term) remain constant across different models. In fact, the VAE in Stable Diffusion is trained following the VQGAN approach, which uses both $L1$ loss and an additional discriminator for training. Therefore, we cannot effectively estimate the reconstruction term since the decoder distribution is not tractable. To simplify further, we omit the VAE encoding process altogether. Specifically, we skip both encoding and decoding through the VAE and treat the latents produced by the VAE as the dataset samples. Under this assumption, we estimate the offset VLB directly in the latent space.

In the latent space, we model the distribution of latent variables that can be decoded into images using the VAE decoder. We denote the samples in latent space as $x$, and the noisy latent at timestep $t$ as $z_t$. The variational lower bound (VLB) in the latent space is given by Kingma et al. (2021):

$$-\log p(x) \le -\text{VLB}(x) = D_{KL}(q(z_1|x)||p(z_1)) + \mathbb{E}_{q(z_0|x)}[-\log p(x|z_0)] + \mathcal{L}_T(x), \tag{13}$$

where the first two terms depend only on the noise schedule, and we treat these terms as irreducible losses since the noise schedule is fixed across all models. The third term is the KL divergence between each pair of the reverse process $p(z_t|z_{t+1})$ and the forward process $q(z_t|x, z_{t+1})$:

$$\mathcal{L}_T(x) = \sum_{i=1}^{T} \mathbb{E}_{q(z_{t(i)}|x)}[D_{KL}(q||p)]. \tag{14}$$

Since we assume that the forward and reverse processes share the same variance and both $p$ and $q$ are Gaussian distributions, the KL terms reduce to weighted $L2$ distances:

$$\mathcal{L}_T(x) = \frac{1}{2}\mathbb{E}_{\epsilon \sim \mathcal{N}(0,I)} \left[ \sum_{i=1}^{T} (SNR(s) - SNR(t))\|x - x_\theta(z_t;t)\|_2^2 \right], \tag{15}$$

where $s = t - 1$. In the limit as $T \to \infty$, the loss becomes:

$$\mathcal{L}_T(x) = -\frac{1}{2}\mathbb{E}_{\epsilon \sim \mathcal{N}(0,I)} \int_0^1 SNR'(t)\|x - x_\theta(z_t;t)\|_2^2 \, dt. \tag{16}$$

In our case, we utilize the velocity $\mathbf{v}$ to predict the clean sample $x$ and compute the VLB.

**Normalizing Flows** Another method to compute the likelihood in our diffusion model is by viewing the diffusion process as a type of normalizing flow. Specifically, we leverage the theoretical results from Neural ODEs, which allow us to connect continuous normalizing flows with the evolution of probability density over time. In Neural ODEs, the transformation of data through the flow can be described by the following differential equation for the state variable $x_t$ as a function of time:

$$\frac{dx_t}{dt} = f_\theta(x_t, t), \tag{17}$$

where $f_\theta(x_t, t)$ represents the network that predicts the time-dependent vector field $\mathbf{v}$. To compute the change in log-probability of the transformed data, the log-likelihood of the input data under the flow is given by:

$$\frac{d \log p(x_t)}{dt} = -\text{Tr}\left( \frac{\partial f(x_t, t)}{\partial x_t} \right), \tag{18}$$

where Tr represents the trace of the Jacobian matrix of $f_\theta(x_t, t)$. This equation describes how the log-density evolves as the data is pushed forward through the flow. To compute the likelihood, we integrate the following expression over the trajectory from the initial state $t_0$ to the terminal state $t_1$:

$$\log p(x_{t_1}) = \log p(x_{t_0}) - \int_{t_0}^{t_1} \text{Tr}\left( \frac{\partial f_\theta(x_t, t)}{\partial x_t} \right) dt. \tag{19}$$

Here, $\log p(x_{t_0})$ represents the log-likelihood of the initial state (often modeled as a Gaussian), and the integral accounts for the change in probability density over time as the data evolves through the ODE. In our formulation, the network predicts the velocity $\mathbf{v}_\theta(x_t, t) = x_t' = \alpha_t' x_0 + \beta_t' \epsilon$, which corresponds to the derivative of $x_t$ with respect to time. Thus, we start with clean samples, estimate the velocity, perform an iterative reverse-time sampling, and convert the samples into Gaussian noise. We can then compute the prior likelihood of the noise easily and add it to the probability shift accumulated during reverse sampling. In our experiments, we set the steps of reverse sampling to 500 to obtain a rather accurate estimation.

## G SCALING FLOPS COUNTING

In this section, we provide a detailed explanation of our FLOPs scaling calculations. Several prior works have employed different methods for counting FLOPs. In Kaplan et al. (2020), the authors exclude embedding matrices, bias terms, and sub-leading terms. Moreover, under their framework, the model dimension $d_{model}$ is much larger than the context length $l_{ctx}$, allowing them to disregard context-dependent terms. Consequently, the FLOPs count $N$ for their model is given by:

$$M = 12 \times d_{model} \times n_{layer} \times (2d_{attn} + d_{ff}), \tag{20}$$

where $d_{model}$ represents the model dimension, $n_{layer}$ denotes the depth of the model, $d_{attn}$ refers to the attention dimension, and $d_{ff}$ represents the feed-forward layer dimension.

In contrast, Hoffmann et al. (2022) includes all training FLOPs, accounting for embedding matrices, attention mechanisms, dense blocks, logits, and context-dependent terms. Specifically, their FLOP computation includes:

- **Embedding**: $2 \times l_{ctx} \times N_{voc} \times d_{model}$
- **Attention**:
  - **QKV Mapping**: $2 \times 3 \times l_{ctx} \times d_{model} \times d_{model}$
  - **QK**: $2 \times l_{ctx} \times l_{ctx} \times d_{model}$
  - **Softmax**: $3 \times n_{head} \times l_{ctx} \times l_{ctx}$
  - **Mask**: $2 \times l_{ctx} \times l_{ctx} \times d_{model}$
  - **Projection**: $2 \times l_{ctx} \times d_{model} \times d_{model}$
- **Dense**: $2 \times l_{ctx} \times (d_{model} \times d_{ff} \times 2)$
- **Logits**: $2 \times l_{ctx} \times d_{model} \times N_{voc}$

Further details can be found in the Appendix **F** of Hoffmann et al. (2022).

In Bi et al. (2024), the authors omit the embedding computation but retain the context-dependent terms, which aligns with our approach. The parameter scaling calculation for vanilla Transformers in this paper follows the same format as theirs. We now present detailed scaling FLOPs calculations for the In-Context Transformers and Cross-Attn Transformers used in our experiments.

Attention blocks are the primary components responsible for scaling in Transformer architectures. In line with previous studies, we only consider the attention blocks, excluding embedding matrices and sub-leading terms. Unlike large language models (LLMs), our model dimension is comparable to the context length, and therefore, we include context-dependent terms. In this section, we present FLOPs per sample rather than parameters, as different tokens participate in different parts of the cross-attention computation. Additionally, since our input length is fixed, the FLOPs per sample are straightforward to compute.

**In-Context Transformers** In-Context Transformers process a joint embedding consisting of text, image, and time tokens, all of which undergo attention computation. Tab. 3 details the FLOPs calculations for a single attention layer.

Table 3: Scaling FLOPs Calculation in In-Context Transformers

| Operation | FLOPs per Sample |
|---|---|
| Self-Attn: QKV Projection | $3 \times 2 \times n_{layer} \times l_{ctx} \times d_{model} \times 3 \times d_{attn}$ |
| Self-Attn: QK | $3 \times 2 \times n_{layer} \times l_{ctx} \times l_{ctx} \times d_{attn}$ |
| Self-Attn: Mask | $3 \times 2 \times n_{layer} \times l_{ctx} \times l_{ctx} \times d_{attn}$ |
| Self-Attn: Projection | $3 \times 2 \times n_{layer} \times l_{ctx} \times d_{model} \times d_{attn}$ |
| Self-Attn: FFN | $3 \times 2 \times 2 \times n_{layer} \times l_{ctx} \times 4 \times d_{model}^2$ |

In our experiments, we set $d_{model} = d_{attn}$, and $l_{ctx} = 377$, where $l_{ctx} = l_{img}(256) + l_{text}(120) + l_{time}(1)$. Thus, the simplified FLOPs-per-sample scaling equation $M$ is:

$$M = 72 \times l_{ctx} \times n_{layer} \times d_{model}^2 + 12 \times n_{layer} \times l_{ctx}^2 \times d_{model} \tag{21}$$

**Cross-Attn Transformers** In Cross-Attn Transformers, each attention block consists of self-attention and cross-attention mechanisms to integrate text information. The cross-attention uses image embeddings as the query and text embeddings as the key and value. The attention mask reflects the cross-modal similarity between image patches and text segments. As a result, the FLOPs calculation differs from that of models using joint text-image embeddings. Tab. 4 lists the FLOPs costs for each operation.

Table 4: Scaling FLOPs Calculation in Cross-Attn Transformers

| Operation | FLOPs per Sample |
|---|---|
| Self-Attn: QKV Projection | $3 \times 2 \times n_{layer} \times l_{img} \times d_{model} \times 3 \times d_{attn}$ |
| Self-Attn: QK | $3 \times 2 \times n_{layer} \times l_{img} \times l_{img} \times d_{attn}$ |
| Self-Attn: Mask | $3 \times 2 \times n_{layer} \times l_{img} \times l_{img} \times d_{attn}$ |
| Self-Attn: Projection | $3 \times 2 \times n_{layer} \times l_{img} \times d_{model} \times d_{attn}$ |
| Cross-Attn QKV | $3 \times 2 \times n_{layer} \times (l_{img} + 2 \times l_{text}) \times d_{model} \times d_{attn}$ |
| Cross-Attn QK | $3 \times 2 \times n_{layer} \times l_{text} \times l_{img} \times d_{attn}$ |
| Cross-Attn Mask | $3 \times 2 \times n_{layer} \times l_{text} \times l_{img} \times d_{attn}$ |
| Cross-Attn Projection | $3 \times 2 \times n_{layer} \times l_{img} \times d_{model} \times d_{attn}$ |
| FFN | $3 \times 2 \times 2 \times n_{layer} \times l_{img} \times 4 \times d_{model}^2$ |

Based on the experimental settings, we can simplify the FLOPs calculation as follows:

$$M = 84 \times n_{layer} \times l_{img} \times d_{model}^2 + 12 \times n_{layer} \times l_{img}^2 \times d_{attn}$$

$$+12 \times n_{layer} \times l_{text} \times d_{model}^2 + 12 \times n_{layer} \times l_{text} \times l_{img} \times d_{model} \tag{22}$$

**Context-Dependent Terms** From the equations above, it is evident that some context-dependent terms, such as $12 \times n_{layer} \times l_{ctx}^2 \times d_{model}$, cannot be omitted. In our experiments, the aspect ratio of Transformers (width/depth=64) is maintained across all model sizes. The context length $l_{ctx}$ is 377 (image: 256, text: 120, time: 1), and $d_{model} = n_{layer} \times 64$. Since $l_{ctx}$ and $d_{model}$ are comparable, the context-dependent terms must be retained.

## H ABLATIONS

### H.1 DIFFUSION FORMULATION

In diffusion models, various formulations for noise schedules, timestep schedules, and prediction objectives have been proposed. These three components are interdependent and require specific tuning to achieve optimal performance. In this paper, we explore several common formulations and conduct ablation studies to identify the best combination in our setting.

Below, we list the candidate formulations used in our ablation study.

**Noise Schedule**

**Discrete Diffusion Models**

**DDPM** Denoising Diffusion Probabilistic Models (DDPM) (Ho et al., 2020) is a discrete-time diffusion model that generates noisy samples via the following formula:

$$x_t = \alpha_t x_0 + \beta_t \epsilon$$

where $\epsilon$ is Gaussian noise, and $\alpha_t$ and $\beta_t$ satisfy $\alpha_t^2 + \beta_t^2 = 1$. Given a sequence of $\sigma_t$, the scaling factor can be defined as:

$$\alpha_t = \sqrt{\prod_{s=0}^{t}(1 - \sigma_t)} \tag{23}$$

In DDPM, $\sigma_t$ follows:

$$\sigma_t = \sigma_0 + \frac{t}{T}(\sigma_T - \sigma_0) \tag{24}$$

**LDM**   Latent Diffusion Models (LDM) (Rombach et al., 2022), as used in Stable Diffusion, is a variant of the DDPM schedule. It is also a variance-preserving formulation, sharing the same structure as DDPM but employing a different noise schedule:

$$\sigma_t = \left( \sqrt{\sigma_0} + \frac{t}{T} \left( \sqrt{\sigma_T} - \sqrt{\sigma_0} \right) \right)^2$$

**Continuous Diffusion Models**

**VP**   Variance Preserving (VP) diffusion (Song et al., 2020) is the continuous counterpart of DDPM, where the noise is added while preserving variance across timesteps. The sampling process is given by:

$$x_t = e^{-\frac{1}{2} \int_0^t \sigma_s ds} x_0 + \sqrt{1 - e^{-\int_0^t \sigma_s ds}} \epsilon \tag{25}$$

where $t \in [0, 1]$.

**Rectified Flow**   Rectified Flow (RF) (Liu et al., 2022; Lipman et al., 2022; Albergo et al., 2023) is another continuous-time formulation, where a straight-line interpolation is defined between the initial sample $x_0$ and the Gaussian noise $\epsilon$. The process is described by:

$$x_t = (1 - t)x_0 + t\epsilon \tag{26}$$

**Prediction Type**

**Noise Prediction ($\epsilon$)**   The network predicts the Gaussian noise $\epsilon \sim \mathcal{N}(0, I)$ added to the samples during the diffusion process.

**Velocity Prediction (v)**   In this formulation, the network predicts the velocity $\mathbf{v}(x_t, t)$, which is defined as the derivative of the noisy sample $x_t$ with respect to time. If the noisy sample $x_t$ is defined by:

$$x_t = \alpha_t x_0 + \beta_t \epsilon \tag{27}$$

The velocity is given by:

$$\mathbf{v}(x_t, t) = x_t' = \alpha_t' x_0 + \beta_t' \epsilon \tag{28}$$

where $\alpha_t'$ and $\beta_t'$ are the derivatives of $\alpha_t$ and $\beta_t$ with respect to timestep $t$.

**Score Prediction (s)**   The network predicts the score function $\mathbf{s}(x, t) = \nabla \log P(x, t)$, which is the gradient of the log-probability density function. The score can be derived as:

$$\mathbf{s}(x, t) = -\frac{\epsilon}{\beta_t} \tag{29}$$

**Timestep Sampling Schedule**

**Uniform Timestep Schedule**   In this schedule, the timestep $t$ is uniformly sampled. For discrete-time diffusion models:

$$t \sim \mathcal{U}(0, 1, 2, \ldots, 999) \tag{30}$$

For continuous-time diffusion models:

$$t \sim \mathcal{U}(0, 1) \tag{31}$$

**Logit-Normal (LN) Timestep Schedule** The Logit-Normal (LN) timestep schedule $\pi_{ln}(t; m, s)$, proposed in Esser et al. (2024), generates timesteps according to the following distribution:

$$\pi_{ln}(t; m, s) = \frac{1}{s\sqrt{2\pi}} \cdot \frac{1}{t(1-t)} \exp\left(-\frac{(logit(t) - m)^2}{2s^2}\right), \tag{32}$$

where $logit(t) = \log\left(\frac{t}{1-t}\right)$. The LN schedule has two parameters: $m$ and $s$. It defines an unimodal distribution, where $m$ controls the center of the distribution in logit space, shifting the emphasis of training samples towards noisier or cleaner regions. The parameter $s$ adjusts the spread of the distribution, determining its width. As suggested in Esser et al. (2024), to obtain a timestep, we first sample $u \sim \mathcal{N}(m, s)$, and then transform it using the logistic function. For discrete-time diffusion, after obtaining $t \sim \pi_{ln}(t; m, s)$, we scale $t$ by $t = \text{round}(t \times 999)$ to obtain a discrete timestep. Following Esser et al. (2024), we utilized the parameters $m = 0.0, s = 1.00$ and didn't sweep over $m$ and $s$. More details and visualizations can be found in Esser et al. (2024) Appendix **B.4**.

We conducted a series of experiments using different combinations of formulations. Selective combinations are listed in Tab. 5. A '$-$' indicates that the combination is either not comparable with other formulations or that training diverges. We assume that the choice of formulation will not be significantly affected by specific model designs or datasets. All experiments were conducted using Pixart (Chen et al., 2023), a popular text-to-image diffusion transformer architecture. Specifically, we used a small model with 12 layers and a hidden size of 384, setting the patch size to 2. The models were trained on JourneyDB (Sun et al., 2024), a medium-sized text-to-image dataset consisting of synthetic images collected from Midjourney. All models were trained for 400k steps using AdamW as the optimizer. As shown in Tab. 5, the optimal combination is **[RF, LN, v]**, achieving an FID of **36.336** and a Clip Score of **0.26684**. This combination achieved the best performance on both metrics and therefore, we used this setting in the remaining experiments.

Table 5: Ablation on diffusion formulations.

| Noise Schedule | Timestep Schedule | Prediction Type | FID | CLIP Score |
|----------------|-------------------|-----------------|-----|------------|
| DDPM | Uniform | $\epsilon$ | 40.469 | 0.26123 |
| DDPM | Uniform | v | 74.049 | 0.23136 |
| DDPM | LN | $\epsilon$ | 40.100 | 0.26283 |
| LDM | Uniform | $\epsilon$ | 39.001 | 0.26423 |
| LDM | Uniform | v | $-$ | $-$ |
| LDM | LN | $\epsilon$ | 38.196 | 0.26624 |
| VP | Uniform | $\epsilon$ | 44.343 | 0.26148 |
| VP | Uniform | v | 41.808 | 0.26320 |
| VP | Uniform | s | 45.808 | 0.25970 |
| VP | LN | $\epsilon$ | 42.872 | 0.26354 |
| VP | LN | v | 44.107 | 0.26380 |
| RF | Uniform | v | 44.840 | 0.25682 |
| RF | Uniform | s | $-$ | $-$ |
| RF | Uniform | $\epsilon$ | $-$ | $-$ |
| RF | LN | v | **36.336** | **0.26684** |

## H.2 EFFECT OF EMA ON LOSS

The Exponential Moving Average (EMA) coefficient plays a crucial role in shaping the loss curve and affects final outcomes. In EMA, the loss $l$ is updated via $l = (1 - \alpha_{\text{EMA}})l + \alpha_{\text{EMA}}v$, where $v$ is the latest loss value. This smoothing mechanism reduces fluctuations in cumulative loss. However, during the early stages of training, EMA can overestimate the loss. As shown in Fig. 7, higher EMA coefficients lead to elevated loss values compared to actual loss, potentially introducing bias in curve fitting and causing inefficient use of compute. From Fig. 7, we observe that $\alpha_{\text{EMA}} = 0.9$ strikes a good balance, effectively smoothing the loss while only mildly inflating values early in training.

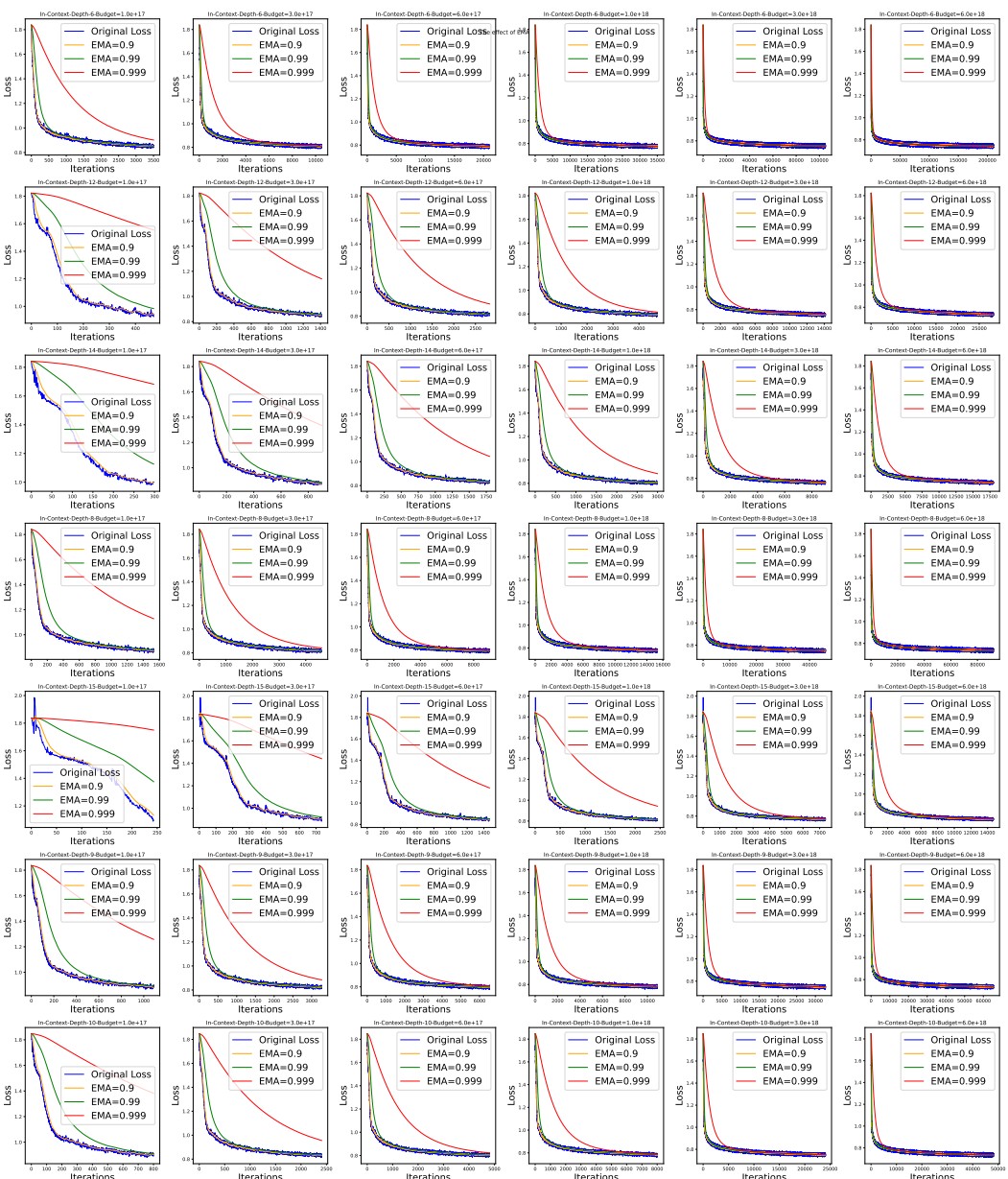

Figure 7: Effect of EMA on loss values.

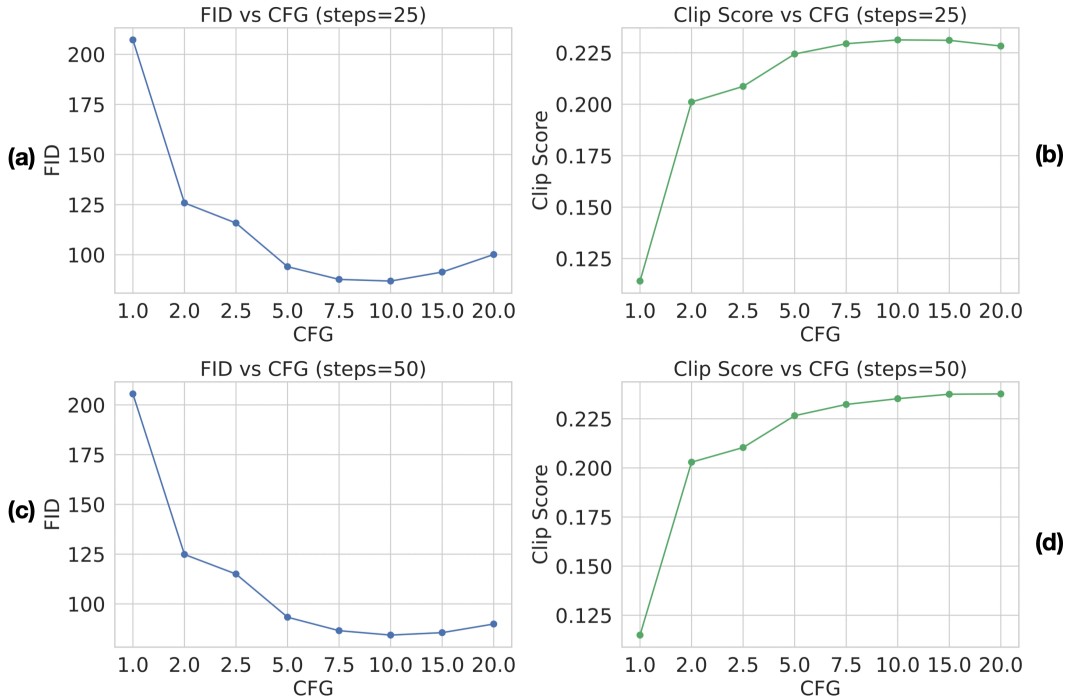

Figure 8: Ablation on sampling steps.

### H.3 CLASSIFIER-FREE GUIDANCE & SAMPLING STEPS

We conduct ablation studies on CFG scales and sampling steps using compute-optimal models trained with a budget of $6 \times 10^{18}$. In Fig. 22, we evaluate various CFG scales (2.5, 5.0, 7.5, 10.0) across different step counts and compute the FID. We find that 25 steps suffice for good performance. Next, fixing the number of steps to 25, we assess different CFG scales. As shown in Fig. 9, a CFG scale of 10.0 yields the best FID and is thus chosen as the default.

Although CFG influences generated results, the linearity of the FID scaling curves remains largely invariant to the CFG scale. To demonstrate this, we plot FID scaling curves under various CFG settings in Fig. 10. The results confirm that scaling behavior remains linear across all tested CFG values.

### H.4 EFFECT OF EMA ON MODEL

Maintaining an EMA copy of the model is a common practice in diffusion models, as it often leads to improved sample quality. To evaluate its impact, we maintain a model copy with an EMA coefficient of 0.99 during training and measure the FID of the EMA versions across different compute budgets. As shown in Fig. 11(a), the EMA models also exhibit a power-law scaling trend.

### H.5 SCALING LAWS FOR MORE ARCHITECTURES & DATA

We extend our scaling law analysis to additional architectures, including PixArt (Chen et al., 2023) and Flux. As shown in Fig. 13(b) and Fig. 12(b), both models exhibit clear linear behavior in their loss curves. These results suggest that scaling laws are consistent across different diffusion transformer architectures.

We further test the applicability of scaling laws on higher-resolution data. Specifically, we train models on $512 \times 512$ images using budgets ranging from `1e17` to `6e18`. The results, presented in Fig. 14(b), show clear linear trends, indicating that scaling laws hold regardless of data resolution.

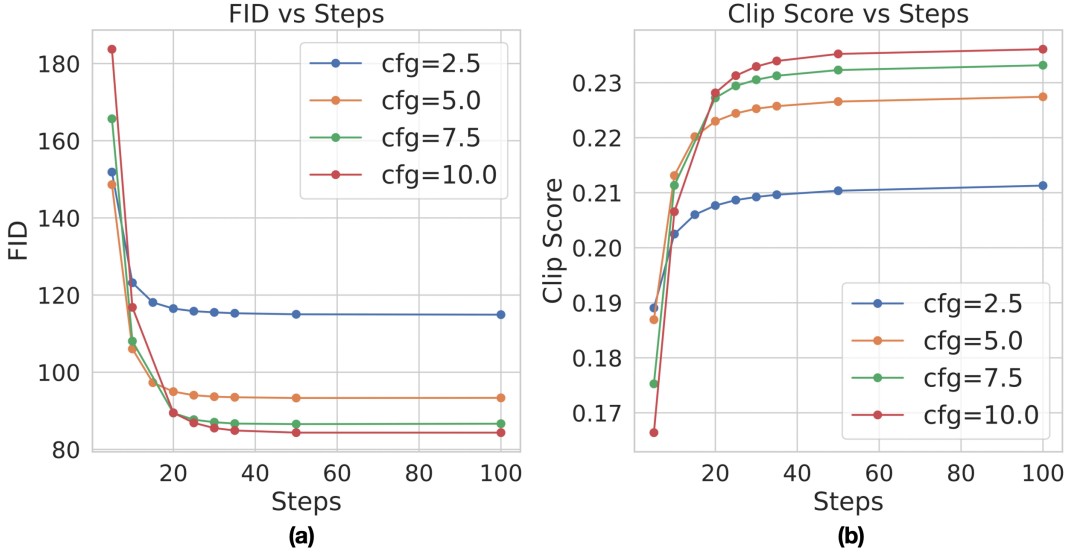

Figure 9: Ablation on CFG scale.

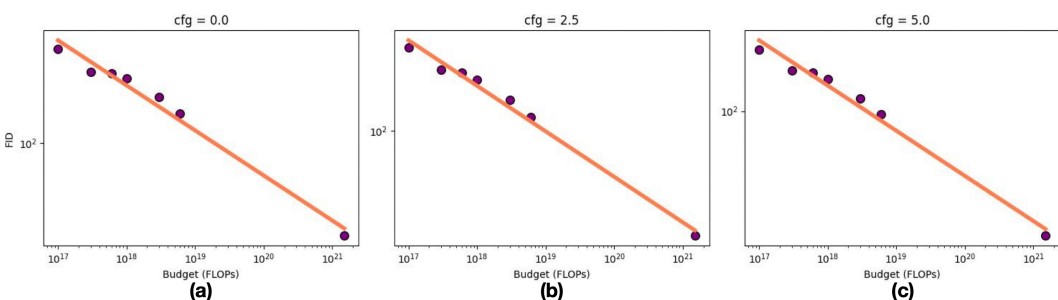

Figure 10: Scaling curves for different CFG scales.

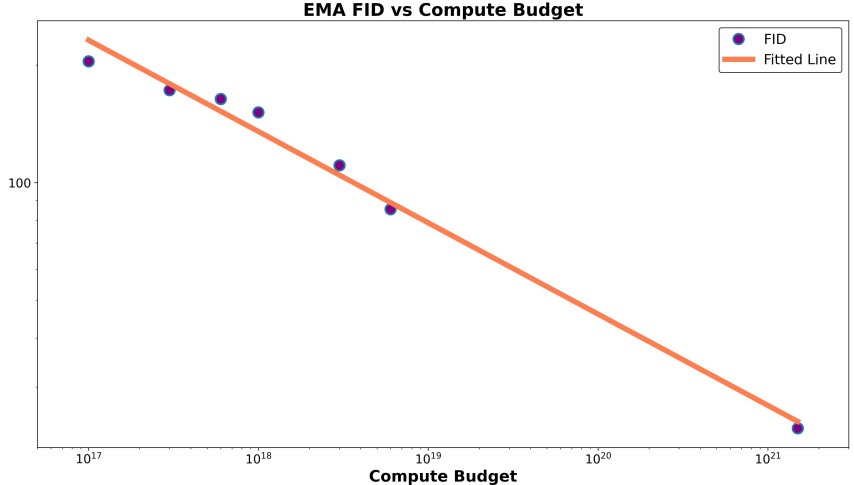

Figure 11: Scaling curves for EMA models.

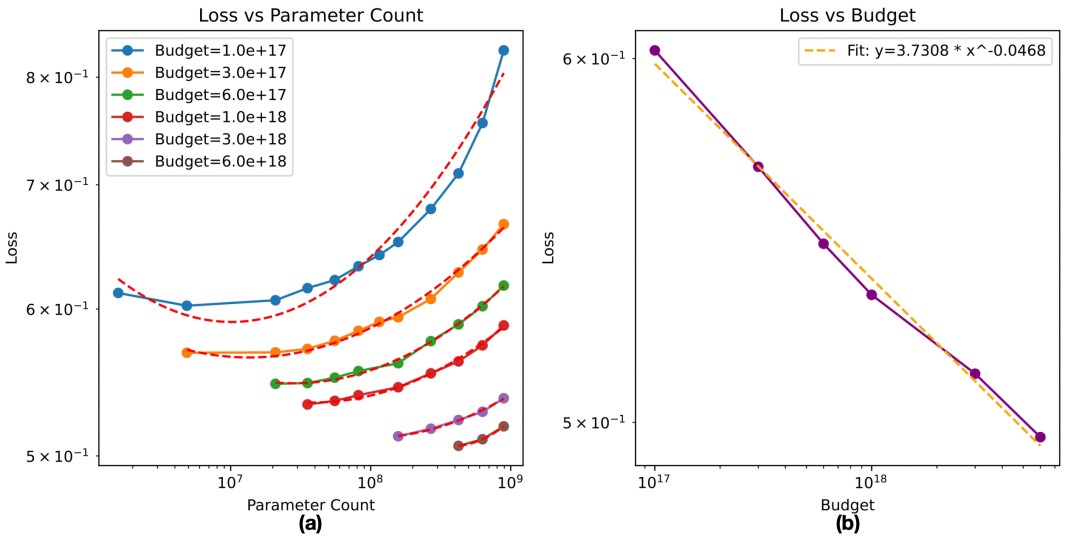

Figure 12: Scaling laws for Flux.

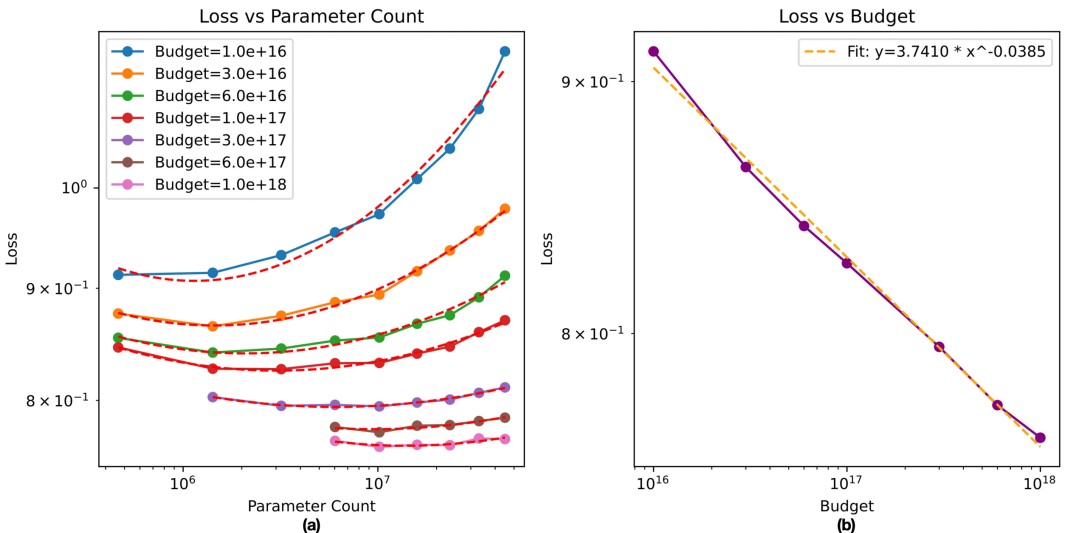

Figure 13: Scaling laws for PixArt.

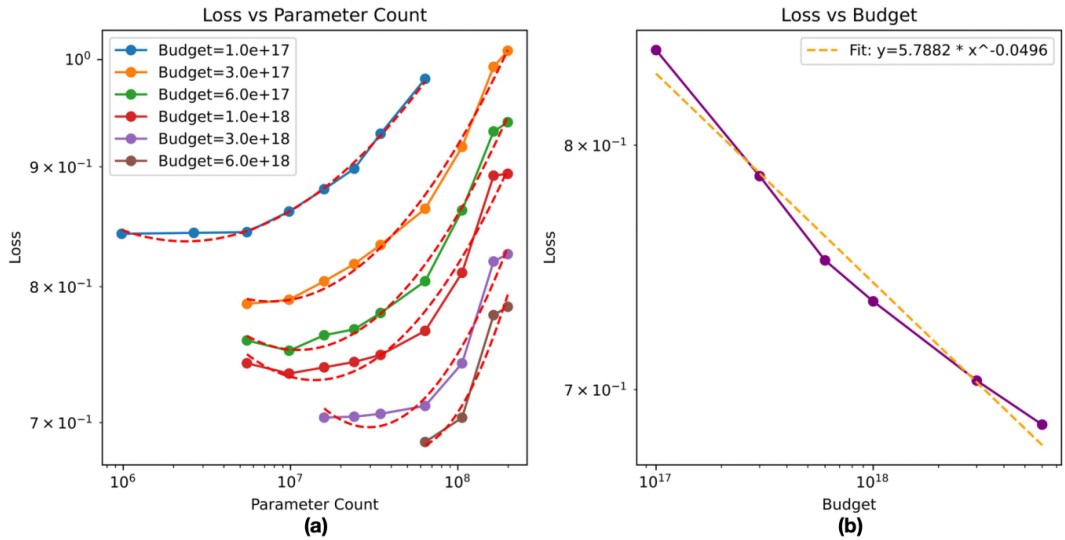

Figure 14: Scaling laws for 512×512 resolution.

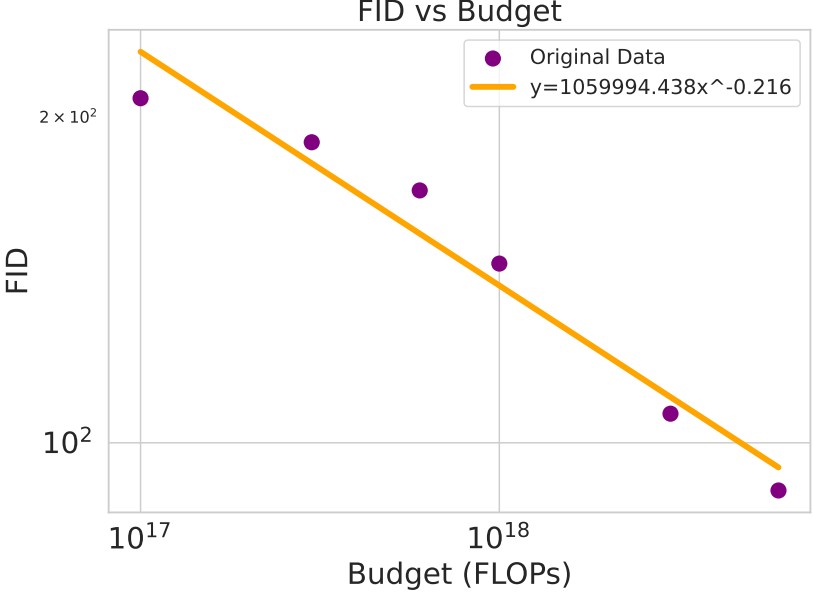

Figure 15: Scaling curves for FID on sparse-captioned dataset.

### H.6 DATA ASSESSMENT

To demonstrate how scaling laws can assess data quality, we apply them to a dataset with the same images as our main experiment but using sparse tag captions instead of dense descriptions. As shown in Fig. 15, the FID scaling exponent is -0.216, whereas the main dataset achieves -0.234. This indicates that dense captions lead to faster FID improvement, suggesting superior data quality compared to sparse tags.

### H.7 TRANSFER RESULTS ON MORE DATASETS

To further validate the generalizability of our scaling laws, we perform transfer experiments on Flickr30k (Plummer et al., 2016) and JourneyDB (Sun et al., 2024), following the same setup as

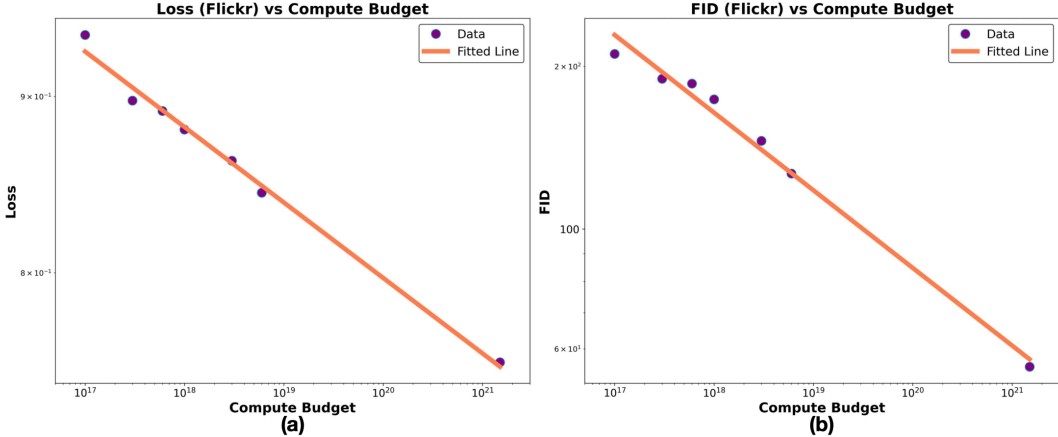

Figure 16: Transfer results of loss and FID on Flickr30k.

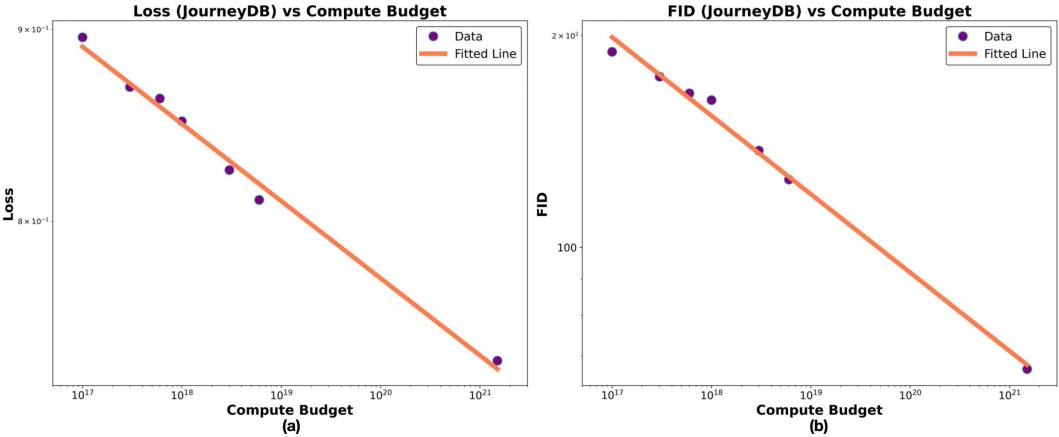

Figure 17: Transfer results of loss and FID on JourneyDB.

with COCO. Models pretrained on our dataset are evaluated on these new domains. As illustrated in Fig. 16 and Fig. 17, both loss and FID maintain clear linear relationships, confirming that our findings transfer well across datasets.

## H.8 GENERATED SAMPLES

We provide qualitative results by showcasing generated samples from models trained with increasing compute. Sampling is performed using fixed prompts and seeds. As shown in Fig. 18, sample quality improves with more compute. However, our goal is not to produce state-of-the-art results, but rather to understand scaling behavior in diffusion transformers. Hence, we did not curate high-quality datasets or optimize for SOTA image generation.

## H.9 SCALING LAWS ON IMAGENET

While our primary experiments assume data-infinite settings, we also test scaling behavior under data-constrained conditions using ImageNet (Deng et al., 2009). For this, we adopt the model from SiT (Ma et al., 2024a) and vary compute from $2 \times 10^{17}$ to $6 \times 10^{18}$. As shown in Fig. 19(c), the loss curves still follow clear power-law trends, indicating that scaling laws also emerge in finite-data regimes.

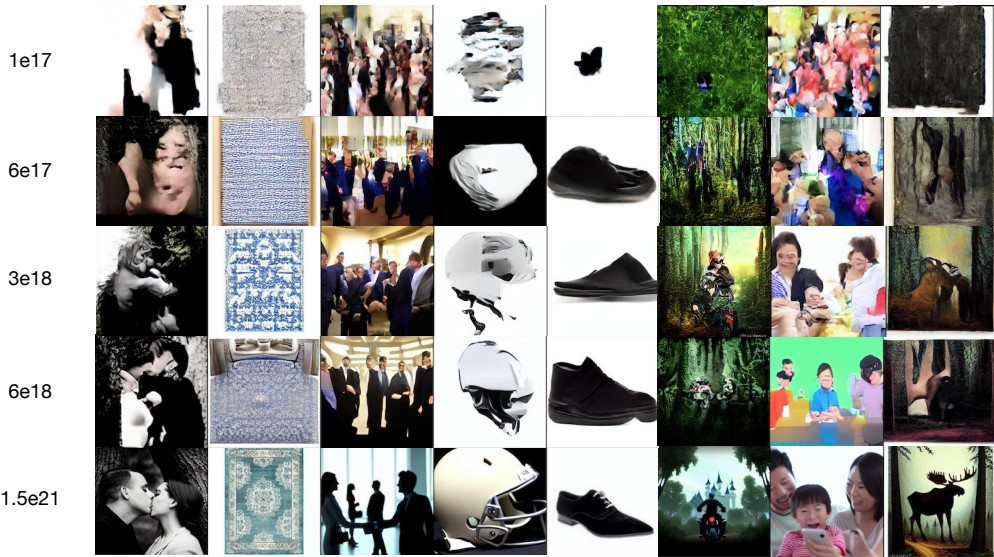

Figure 18: Generated samples from models with increasing compute.

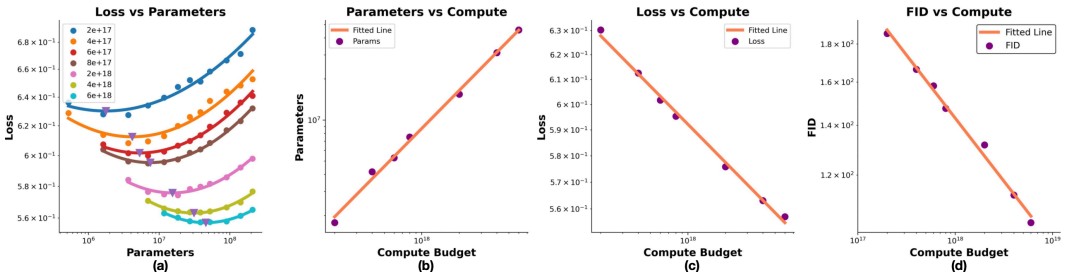

Figure 19: Scaling curves on ImageNet.

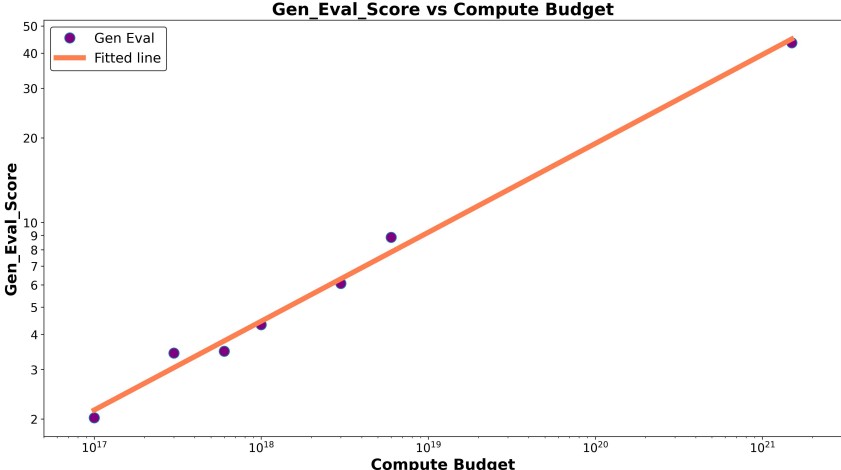

Figure 20: Scaling laws on GenEval benchmark.

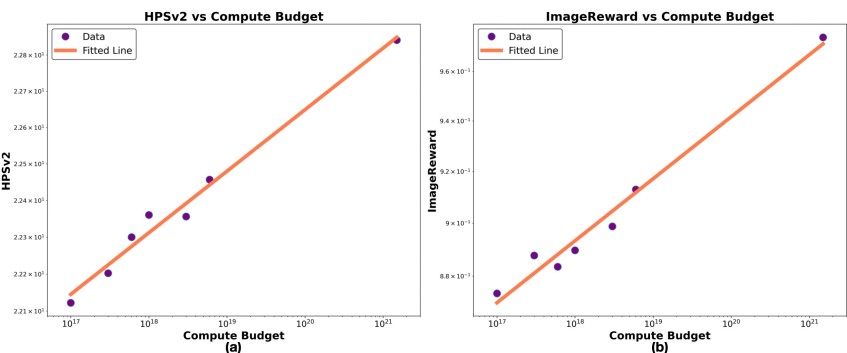

Figure 21: Scaling laws on Human Preference Rewards.

## H.10   GENEVAL RESULTS

In addition to traditional metrics like FID and loss, we evaluate models using the GenEval (Ghosh et al., 2023) benchmark, which measures text-image alignment across aspects such as object count, color, and position. For each compute budget, we evaluate the corresponding compute-optimal model. Results are presented in Fig. 20, further supporting the presence of consistent scaling behavior across diverse evaluation metrics.

## H.11   HUMAN PREFERENCE RESULTS

We also employ human-preference-based reward models, including HPSv2.1 (Wu et al., 2023) and ImageReward (Xu et al., 2023), to evaluate our models. These benchmarks capture alignment with human preference. Across different compute budgets, the performance measured by these human preference rewards also follows a power-law scaling trend. The results are plotted in Fig.21.

## H.12   SAMPLING STEPS

To study the effect of the sampling schedule on scaling behavior, we additionally vary the number of diffusion sampling steps while fixing the classifier-free guidance scale to 10.0. Specifically, we compare models evaluated with 35 and 50 sampling steps. As shown in Figure 22, changing the number of sampling steps shifts the overall performance, but the resulting points still lie on a clear straight line in log–log space. This indicates that the power-law relationship between compute and generation quality remains intact, and that modifying the sampling steps mainly affects the offset of the scaling curve rather than the existence of the scaling law itself.

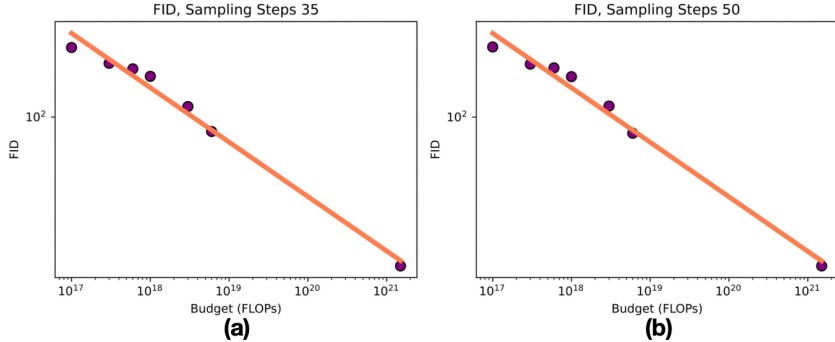

Figure 22: Scaling laws of FID with differnet sampling steps.

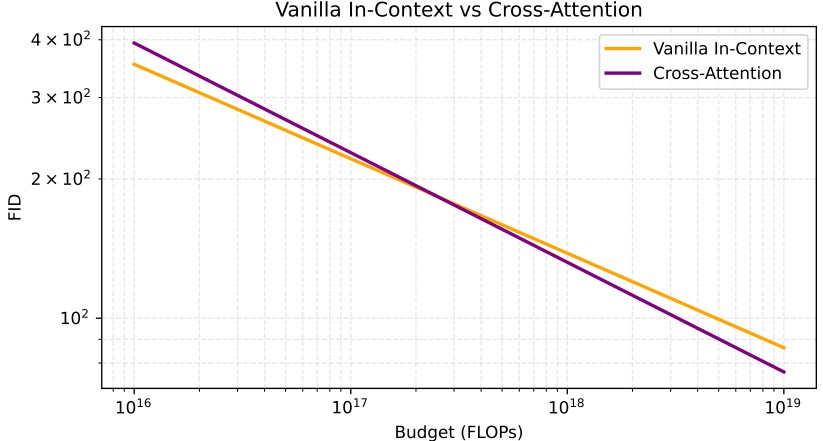

Figure 23: FID results of cross attention vs vanilla in-context.

### H.13 FID RESULTS OF CROSS-ATTN VS VANILLA IN-CONTEXT

We provide the FID results of cross attention architecture and vanilla in-context architecture. The result is shown in Fig. 23, the cross attention architecture has more steep slope and shows greater scalability, which is consistent with the loss curve.

## I LIMITATIONS AND FUTURE DIRECTIONS

**Limited modalities and downstream tasks.** Our empirical study focuses exclusively on the text-to-image setting. We do not experiment with additional modalities such as text-to-video or other multi-modal generative tasks. Moreover, our evaluation is restricted to the core generative objective and a small set of aggregate metrics; we do not consider richer downstream applications such as image editing, inpainting, or other task-specific evaluations. As a result, the extent to which the observed scaling laws transfer to these broader modalities and downstream tasks remains an open question.

**Compute-limited regime only.** All experiments are conducted in a compute-limited regime, where we assume access to sufficiently large datasets and primarily vary the compute budget via model size and training steps. In practice, large-scale training often becomes data-constrained when scaling up compute, and the effective scaling behavior in such regimes may deviate from the laws observed here. In particular, data-constrained settings typically require corrections to standard compute-optimal scaling laws to account for finite-data effects. We leave a systematic study of scaling behavior and its correction under data-limited regimes to future work.

**More accurate hyperparameters.** Our results are obtained using a single, well-tuned set of training hyperparameters for each model family, which are reused across different compute budgets. While this choice reflects a realistic training protocol, it also means that our scaling laws may not reflect fully optimized performance at each operating point. Better or more carefully adapted hyperparameters (e.g., learning rate schedules, regularization strength, or optimization settings) could further reduce the variance and systematic bias in the observed scaling curves, thereby improving the accuracy of compute-optimal predictions. However, with our current setting, we can already reveal the scaling dynamics of diffusion models, and we leave the more accurate hyperparameter as an important direction for future research.

## J    DISCUSSION: IN-CONTEXT VS. CROSS-ATTENTION TRANSFORMERS

Our main experiments in Sec. 4 indicate that, under our controlled setting, cross-attention transformers exhibit a more favorable scaling trend with compute compared to the simple in-context baseline we study. At the same time, recent text-to-image systems, such as MMDiT-style architectures and Flux-like models, report very strong absolute performance with in-context conditioning. This may appear to be in tension with our findings.

We emphasize that this apparent discrepancy can arise from many factors beyond the conditioning mechanism itself. Publicly available systems differ not only in whether they use in-context or cross-attention conditioning, but also in total compute, data mixture and quality, and overall training recipe, all of which strongly affect both performance and observed scaling behavior. In practice, models like Flux are among the strongest systems largely because of their full recipe (data + compute + architecture), rather than solely because in-context conditioning is intrinsically superior to cross-attention. Under the same data and compute budget, a carefully tuned cross-attention architecture could plausibly achieve similar or even better performance. However, such a direct, controlled comparison is not currently available in the literature.

Moreover, modern in-context architectures typically incorporate several important improvements relative to the simple in-context baseline considered in this work: (i) *Stronger conditioning*: they often use multiple CLIP encoders plus a large language model (e.g., T5-XXL) and inject text features into every block via MLPs that modulate attention and FFN layers, leading to much stronger conditioning; (ii) *Better timestep conditioning*: instead of encoding timesteps as in-context tokens, they employ adaptive LayerNorm-style conditioning, where a timestep embedding is mapped to scale/shift/gate parameters that modulate each block, which is empirically more efficient; (iii) *More efficient modality interaction*: as in SD3-style designs, text and image tokens may maintain separate parameter sets, with attention used only for cross-modal interaction and text injected primarily in earlier blocks, reducing cross-modal compute; (iv) *More advanced architectural and training choices*: improved positional encodings, noise/timestep schedules, and other refinements further enhance scaling efficiency.

Taken together, these design choices, combined with different data and compute budgets, can explain why state-of-the-art in-context models achieve very strong performance in practice, without implying a direct, controlled advantage of in-context conditioning over cross-attention in terms of intrinsic scalability. Our results should therefore be interpreted as characterizing scaling behavior within a simplified and carefully matched setting. Extending this analysis to more advanced in-context and cross-attention architectures is an important direction for future work.

