# OpenReview forum: "Scaling Laws for Diffusion Transformers"
_ICLR.cc/2026/Conference — ICLR 2026 Poster_

### Official Review · Reviewer_fsKy · 2025-10-28

**Soundness:** 4
**Presentation:** 3
**Contribution:** 4
**Rating:** 6
**Confidence:** 3

**Summary:**

This paper presents the first systematic investigation of the scaling laws of Diffusion Transformers (DiT) for text-to-image synthesis. The authors conduct extensive expriements accross compute budgets $C$ from $1e17$ to $6e18$, and predict the performance for a larger 1B-model with compute budgets $1.5e21$.

The key contributions are:
1. **First Establishment of Scaling Laws of DiT**: This paper is the first to investigate and confirm the power-law relationship with the compute budget.
1. **Fit the Power Law Equations**: This papers fits the power-law equations for model/data size: $N_{opt}=0.0009\cdot C^{0.5681}$ and $N_{opt}=186.8535\cdot C^{0.4319}$, as well as for the loss $L=3.3943\cdot C^{-0.0273}$.
1. **Validation on Larger Compute Budget**: The derived laws are validated by extrapolating to a significantly larger compute budget ( $1.5e21$), a 1B parameter model, and demonstrating that its loss and performance metrics match the predictions.
1. **Evalution Metrics Justification**: The authors also verify generative quality metrics like FID also follows the scaling laws ($FID = 2.2566 \times 10^6 \cdot C^{-0.234}$), as well as other evaluation metrics like VLB, exact likelihood, GenEval, and human preference scores. This also justifys that these evaluation metrics are suitable for text-to-image synthesis evaluation.
1. **Generative Robustness**: The paper demonstrates the robustness of these laws by showing they hold for out-of-domain (OOD) data (e.g., COCO) and across different model architectures.
1. **Guidance on Model/Data design**: This paper shows difference model architecture (In-Context *v.s.* Cross-Attention) has different coefficients on the power-law equations, which helps evalutate future model designs using small compute budgets.

**Strengths:**

1. This paper for the first time provide a systematic investigation and verification of predictive, quantitative scaling laws of Diffusion Transformer (DiT).
1. The expriments conducted to investigate the scaling laws of DiT is reasonable and thorough, which include large-scale validation, evaluation of various metrics and coefficents comparision of different models.
1. The proposal to use scaling exponents as a predictable benchmark is practically usable and provides the community with a powerful and low-cost tool for architectural and data-quality comparisons.

**Weaknesses:**

The paper is generally good, but there's some mirror "weaknesses" or improvement sugesstion. I am happy to increase the score if they are addressed.
1. **Contradiction in Section 4**: There is a small contradiction between the text and Table 1 when comparing model architectures. The text states, "The Cross-Attention Transformer exhibits a *larger* model exponent". However, Table 1 shows the Cross-Attention model exponent (0.54) is smaller than the Vanilla In-Context model's exponent (0.56). The text's conclusion—that "more resources should be allocated toward scaling the dataset" —is correct and does align with the table (data exponent 0.46 > 0.43).
1. **Some writtings are confusing**:
    1. Line 241, $C$ is the symbol of "Compute", but later it's referenced as compute budget in line 252. First-time reader might not know "Compute" and "Compute budget" are the same.
    1. There are three sub-figures in Figure 1, but they are refered as a whole, e.g., in Line 265, it should refer to the first sub-figure of Figure 1.
    1. The logic from Line 243 to 251 is not clear to me. It says $C=6ND$, a linear relationship between $C$ and $N$ or $D$, but later it ypothesizes $N_{opt} \propto C^{a}$ and $D_{opt} \propto C^{b}$. It's unclear to me why, untill I review Figure 1 that the x-axis is in log-scale and the fitting curve is a strage line. But still the y-axis of Figure 1 does not say they are $N_{opt}$ and $D_{opt}$ so it's still unclear to me how the hypothesis comes from.
    1. Figure 3 shows legend the second and the third sub-fogures of Figure 1 do not. And the second sub-figure of Figure 3's legend is *fitter* curve.

**Questions:**

1. As shown in Figure 3, as the $C$ increases, FID and Loss decrases, and the fitted curve is strage line (as y-axis is log-scale). However we cannot increase compute budget forever to obtain a minus loss, there should be saturation or even overfit. How the scaling curves show staturation and overfit?
1. For a fixed compute budge $C$, the relationship between it and $N$, $C$ is defined as $C=6ND$, but how about $N_{opt}$ and $D_{opt}$?  Say we have a designed a model, and fit the coefficents of $N_{opt}$ and $D_{opt}$, and we have limited compute budget, how can we balance the between $N_{opt}$ and $D_{opt}$?
1. The second and the third sub-figures of Figure 1, the y-axis should be *optimal* parameter $N_{opt}$ and *optimal* token $D_{opt}$?
1. Any experimental FID result of Vanilla In-Context vs Cross-Attention?

---

> ### Author Response · Authors · 2025-11-26
>
> **Q1**: Contradiction in Section 4:
>
> **A1**: We thank the reviewer for pointing out. This is a miswording. The correct description should be “The Cross-Attention Transformer exhibits a smaller model exponent”. We have fixed this in the paper.
>
> **Q2**: Line 241, is the symbol of "Compute", but later it's referenced as compute budget in line 252. First-time reader might not know "Compute" and "Compute budget" are the same.
>
> **A2**: We agree that the current wording can be confusing for first-time readers, since “C” is first introduced as “Compute” and later referred to as “compute budget”. In the revision, we now consistently use the phrase “compute budget C” when the symbol appears, and we clarify explicitly that “Compute” and “compute budget” denote the same quantity.
>
> **Q3**: There are three sub-figures in Figure 1, but they are refered as a whole, e.g., in Line 265, it should refer to the first sub-figure of Figure 1.
>
> **A3**: We also agree that referring to all three panels collectively can make it unclear which sub-figure is being discussed at specific points in the text. In the revised version, we now refer to the relevant sub-figures explicitly whenever we discuss a particular panel, which we believe improves readability and removes this ambiguity.
>
> **Q4**: Figure 3 shows legend the second and the third sub-figures of Figure 1 do not. And the second sub-figure of Figure 3's legend is fitter curve.
>
> **A4**: We thank the reviewer for pointing out the inconsistency in the legends. We have revised these two points in the current version. We have made the legends consistent across the sub-figures: the second and third sub-figures of Figure 1 now include legends, and we have corrected the wording “fitter curve” in Figure 3 to “fitted curve.”

---

> ### Author Response · Authors · 2025-11-26
>
> **Q5**: Overfit and Saturation.
>
> **A5**:  This is an insightful question. We want to respond from two perspectives.
>
> **(1) Saturation/Overfitting.**
>
> Our current scaling curves in Figure 3 are not intended to and cannot describe the regime where training has saturated or overfitted. They are fit under the usual data-abundant / effectively infinite-data assumption used in scaling-law work: the dataset is large enough that each training example is seen only a small number of times, so the regime is essentially compute-limited rather than data-limited. In this setting, the curves follow a clean power law and do not yet show the flattening or upturn that would indicate saturation or overfitting on a validation metric such as FID.
> Under this infinite-data view, the behavior is captured by the asymptotics of the power law. A fit of the form
>  $L(C) = a C^{-b}$
>  tends to 0 as $C \to \infty$ but never becomes negative. Formally, there is no hard “floor”. However, when (b) is small, each additional order of magnitude of compute yields only a tiny improvement. In practice, this manifests as very strong diminishing returns rather than a true negative or zero loss. This is the regime our curves are modeling.
>
> The situation changes in the data-constrained / repeated-data regime studied in prior work [1], where the total dataset is fixed, and the same examples are revisited many times. When the number of repetitions (epochs) is small, the metric continues to improve and behaves similarly to the unique-data / infinite-data case. As repetitions increase, gains from extra compute become diminishing, and the curve departs from the “unique-data” power law. Effectively, one enters a separate data-limited scaling regime with a smaller exponent, which can be viewed as saturation or overfitting behavior.
> In our experiments, we are firmly in the compute-limited, effectively infinite-data regime and do not reach the high-epoch, data-limited setting where such deviations would become visible.
>
> Therefore, our present curves cannot directly show the saturation/overfitting behavior raised by the reviewer. However, a systematic study of data-constrained scaling for DiTs, where the infinite-data assumption is relaxed, and the curve bends away from the unique-data law, is highly relevant for industrial practice, and we think it is important. But this is beyond our current research scope and we leave it to future work.
>
> **(2) Practical role of the scaling laws.**
>
> Finally, we emphasize that our goal is not to claim that the fitted power law holds up to arbitrarily large compute (e.g., to negative loss), but to provide a local description that is useful for practical decisions. Within the realistic compute range of interest, the scaling laws mainly help (i) allocate resources, by predicting how to trade off model size and data/compute to reach a target budget, and (ii) compare architectures and data mixtures, as in Section 4, by examining which design attains better exponents and coefficients. For these purposes, understanding the behavior in the realistic, compute-limited regime, where our curves are approximately straight on a log–log plot, is sufficient.
>
>
> **Q6**: The second and the third sub-figures of Figure 1, the y-axis should be optimal parameter N_{opt} and optimal token D_{opt}?
>
> **A6**: Yes. We have modified the figures to make it more clear .
>
> **Q7**: Experimental FID result of Vanilla In-Context vs Cross-Attention?
>
> **A7**: Yes. We have included the results in the Appendix H.13. As shown in the Figure 23, the FID results have the same trend as the loss, suggesting that our findings are consistent and robust.

---

> ### Author Response · Authors · 2025-11-26
>
> **Q8**: Relation and logic of C=6ND, N_{opt}, and D_{opt}?
>
> **A8**:
>
> We thank the reviewer for the thoughtful questions about the relationship between (C = 6ND) and ($N_{opt}, D_{opt}$). To make this fully transparent, we organize our response into two parts:
>  - why $C \approx 6ND$
>  - how we determine (and implicitly trade off) $N_{opt}$ and $D_{opt}$ for a fixed compute budget;
>
> **(1) Why $C \approx 6ND$.**
>
> First, $C \approx 6ND$ is not a definition that we impose on (C), (N), and (D). Instead, it is a standard approximation to the total training FLOPs of a Transformer model (see, e.g., Sec. 2.1 of [2]). Here (C) denotes the total number of floating–point operations (additions and multiplications) performed over the entire training run.
>
> Almost all computation in a Transformer comes from matrix multiplications and elementwise operations **inside the Transformer blocks**, so (C) can be obtained by summing the FLOPs of all operations in the forward and backward passes. Embedding layers and the final output projection contain relatively few parameters, and their FLOPs do not grow with model size, so they are typically ignored in this estimate.
>
> Consider a single attention layer. For one token with input dimension $d_{model}$ and attention dimension $d_{attn}$, each of the (Q), (K), and (V) projections performs a matrix multiplication of shape $(1 \times d_{model}) \times (d_{model} \times d_{attn}))$, which costs $2d_{model}d_{attn}$ FLOPs. Since there are three such projections, the QKV cost is $3 \times 2d_{model} d_{attn})$ FLOPs per token. Applying the same reasoning to attention, FFN, and other sublayers and summing over all $n_{layer}$ Transformer blocks, we find that a single forward pass of one token through all blocks costs approximately
>  $$
>  C_{forward}
>  \approx
>  4d_{model} n_{layer}\bigl(2 d_{attn} + d_{ff}\bigr),
>  $$
>  while the number of parameters in these blocks is
>  $$
>  N \approx
> 2 d_{model} n_{layer}\bigl(2 d_{attn} + d_{ff}\bigr).
>  $$
>  Thus $C_{forward} \approx 2N$.
>
> It is standard to assume that the backward pass is about twice as expensive as the forward pass, so the FLOPs for one optimization step and one token are
>  $$
>  C_{step,token}
>  \approx
>  2N (forward) + 4N (backward)
>  = 6N.
> $$
> Now consider a training step with batch size (B) and sequence length (S), so there are (BS) tokens per step. The FLOPs per step are
>  $$
>  C_{step} \approx 6NBS.
>  $$
>  Over the full training run, the model processes a total of (D) tokens, where $$D = steps \times BS$$ is exactly the total number of tokens seen during training. The total training compute therefore satisfies
>  $$
>  C \approx 6 N D.
>  $$
> This is the origin of (C = 6ND): it is an approximate FLOPs formula derived by explicitly counting floating–point operations in the Transformer, not a constraint that we arbitrarily impose on (C), (N), and (D).

---

> ### Author Response · Authors · 2025-11-26
>
> **(2) How we determine and “trade off” $N_{opt}$ and $D_{opt}$.**
>
> The $N_{opt}$ and $D_{opt}$ actually correspond to the lowest point of each parabola. These two optimal refer to the same configuration. So there is no need to trade off. We will explain and verify why. Following [3], we view the final loss as a function of model size and data size, (L(N,D)), and define the compute–constrained optimum as
>  $$
>  N_{opt}(C),D_{opt}(C)
>  = \arg\min_{N,D;s.t.\mathrm{FLOPs}(N,D)=C} L(N,D).
>  $$
>  For a fixed compute budget (C), this optimization yields a single best pair $(N_{opt}(C), D_{opt}(C))$. In this sense, the trade-off between model size and data size is already encoded in the constraint $\mathrm{FLOPs}(N, D)=C$.
>
> Empirically, this procedure is exactly what is visualized in Figure 1 (left). For each compute budget $C \in {10^{17}, 3\times10^{17}, \ldots}$, we choose a discrete set of n model sizes $N^{\(k\)}$, $k=1,2,3,..., n$  (implemented by varying the number of Transformer blocks and the hidden dimension). For each $N^{\(k\)}$, the total number of training tokens is fixed by the compute constraint: using $C \approx 6ND$ , we set
>  $
>  D^{\(k\)} \approx \frac{C}{6N^{\(k\)}}.
>  $
>
> We then train a separate model for each $\(N^{\(k\)}, D^{\(k\)}\)$ and record its final validation loss. For a given (C), these losses as a function of $N^{\(k\)}$ produce the U-shaped curves (parabolas) in Figure 1 (left). We fit a quadratic to these points and take its minimum. The parameter count at this minimum is $N_{opt}(C)$, and the corresponding token count is
>  $
>  D_{opt}(C) \approx \frac{C}{6N_{opt}(C)}.
>  $
>
> Thus, for each budget (C), the pair $(N_{opt}(C), D_{opt}(C))$ is exactly the coordinate of the lowest (purple) point on the corresponding parabola in Figure 1 (left): it gives the model size and total number of tokens for the best-performing run under that compute budget. The middle and right subfigures of Figure 1 then plot these optimal values $N_{opt}(C)$ and $D_{opt}(C)$ as functions of C.
>
> From the construction above, for every compute budget (C) we have
>  $$
>  \mathrm{FLOPs}\bigl(N_{opt}(C), D_{opt}(C)\bigr)
>  \approx C
>  \quad\Rightarrow\quad
>  6N_{opt}(C)D_{opt}(C) \approx C.
>  $$
> Then let's check this in our paper. In Figure 1 (middle and right) we take the empirically obtained $N_{opt}(C)$ and $D_{opt}(C)$ and fit simple power laws
>  $$
>  N_{opt}(C) = \alpha C^{a}, \qquad
>  D_{opt}(C) = \beta C^{b}.
>  $$
> In our experiments these fits yield
>  $$
>  N_{opt}(C) = 0.0009C^{0.5681}, \qquad
>  D_{opt}(C) = 186.8535C^{0.4319}.
>  $$
>  The exponents satisfy
>  $$
>  a + b = 0.5681 + 0.4319 \approx 1.
>  $$
>
> Plugging the power laws into the compute relation gives
>  $$
>  6N_{opt}(C)D_{opt}(C)
>  \approx 6 \times 0.0009 \times 186.8535 \times C^{0.5681 + 0.4319}
>  \approx C,
>  $$
>  which is proportional to (C), as required by $C \approx 6ND$. In other words, the empirical result $a + b \approx 1$ is exactly what we expect from the underlying compute constraint: it confirms that our fitted scaling laws for $N_{opt}(C)$ and $D_{opt}(C)$ are internally consistent with the FLOPs accounting expressed by $C \approx 6ND$.
>
> [1] Scaling Data-Constrained Language Models. Muennighoff et al. https://arxiv.org/pdf/2305.16264
>
> [2] Scaling Laws for Neural Language Models. Kaplan et al. https://arxiv.org/pdf/2001.08361
>
> [3] Training Compute-Optimal Large Language Models. Hoffmann et al.  https://arxiv.org/pdf/2203.15556

---

> > ### Comment · Reviewer_fsKy · 2025-11-27
> >
> > Thanks for your replies. My concerns are addressed. I am going to raise my score. Please also add your explanations to the submission if needed. Also, please add an explanation of how the form of Eq (6) is determined (e.g., Figure 1 shows the fitting curve is a straight line, and the x-axis is in log-scale).

---

> > > ### Author Response · Authors · 2025-11-27
> > >
> > > We thank the reviewer for the recognition and helpful advice. We have updated the manuscript accordingly, and all changes are highlighted in **blue**. Concretely:
> > > - We add an explanation for the form of Eq. (6) (line 248).
> > >
> > > - We replace “compute” with “compute budget” and explicitly clarify that these two terms are used interchangeably (line 235).
> > >
> > > - We correct the references to the subfigures in Figure 1 (lines 290 and 294).
> > >
> > > - We add a limitations and future directions section to discuss overfitting and saturation (around line 1805).
> > >
> > > - We include an FID comparison between vanilla in-context transformers and cross-attention transformers (around line 1767).
> > >
> > > - We provide a clearer discussion of the relation $C = 6ND$ and how to get the $N_{opt}$ and $D_{opt}$ (line 238, 247).
> > >
> > > - We revise the wording related to the model exponent (line 476).

---

> > > ### Comment · Reviewer_fsKy · 2025-11-28
> > >
> > > Hi, I recommend changing
> > >
> > > > first subfigure of Fig. 1
> > >
> > > to
> > >
> > > > Fig. 1(a)

---

> > > > ### Author Response · Authors · 2025-11-28
> > > >
> > > > Dear Reviewer,
> > > >
> > > > Thank you for your helpful comment. We have revised the manuscript accordingly. In Figure 1, we have added the labels “(a)”, “(b)”, and “(c)” to the three sub-figures. In the main text, the phrase “first sub-figure of Fig. 1” has been replaced with “Fig. 1(a)” to ensure consistency between the figure and the manuscript.
> > > >
> > > > Best regards,
> > > >
> > > > The authors

---

> > > > > ### Comment · Reviewer_fsKy · 2025-11-28
> > > > >
> > > > > Please do this for all figures as well.
> > > > >
> > > > > I don't have more concerns.
> > > > >
> > > > > I want to raise my score from 6 to 8, but I don't think I can anymore...

---

> > > > > > ### Author Response · Authors · 2025-11-28
> > > > > >
> > > > > > Dear Reviewer,
> > > > > >
> > > > > > Following your suggestion, we have added labels “(a)”, “(b)”, “(c)”, etc., to all figures that contain multiple sub-figures. We have also updated all corresponding references in the main text to match these labels and ensure consistency throughout the manuscript.
> > > > > >
> > > > > > Best regards,
> > > > > >
> > > > > > The authors

---

### Official Review · Reviewer_y2Z9 · 2025-10-30

**Soundness:** 3
**Presentation:** 3
**Contribution:** 3
**Rating:** 6
**Confidence:** 3

**Summary:**

The paper investigates the scaling laws of text-to-image diffusion transformers (DiTs). It shows that these models follow a scaling behavior similar to that of large language models: the training loss and other performance metrics exhibit a power-law relationship with compute when the model size and token count are optimally balanced. Furthermore, the paper demonstrates that this trend generalizes across out-of-domain datasets and different model architectures.

**Strengths:**

* The paper is very well written.

* The findings enable practitioners to tune the hyperparameters of DiTs more efficiently.

* The paper demonstrates scaling laws not only with respect to the loss, but also for other useful metrics such as FID and human preference reward.

**Weaknesses:**

Beyond text-to-image generation, diffusion models have also been applied to tasks such as class-conditioned image generation and text-to-video generation. However, the paper only experiments on text-to-image generation, so it remains unclear whether the same scaling laws extend to these other tasks.

**Questions:**

Overall, I enjoyed reading this paper. While the techniques are not new and the results are not particularly surprising given the established scaling laws of large language models, confirming that similar laws hold for DiTs is nevertheless a valuable and solid contribution. I have a few questions listed below and hope the authors can address them in the rebuttal.

* I would like to confirm how Figure 2 is derived. Is it obtained using the same procedure as Figure 1? That is, for each metric, (1) train models of varying sizes under different compute budgets, (2) fit a parabola between the metric and model size for each compute budget, (3) identify the optimal model size for each compute budget, and (4) plot the metric versus compute budget using these optimal model sizes?

* Eq. 4: This equation is somewhat confusing. First, N is not defined. Second, the summation index i does not appear in the summand, making it unclear what is being summed over.

* In Section 3.1, "Variational Lower Bound" and "Exact Likelihood" are subpoints under "Likelihood." However, in the current formatting, "Likelihood," "Variational Lower Bound," and "Exact Likelihood" appear at the same heading level, which may mislead readers into thinking they are parallel sections, causing potential confusion.

* Figure 5: How many data points are included in each line?

* Section 4 states that the experiments show cross-attention transformers exhibit a superior scaling trend compared to in-context transformers. However, as discussed in Section 4, recent works suggest that in-context conditioning performs better. Could the authors clarify where this discrepancy might come from?

---

> ### Author Response · Authors · 2025-11-26
> **Rebuttal 1**
>
> **Q1**: Scaling laws for class-conditioned image generation and text-to-video generation.
>
> **A1**: We thank the reviewer for raising the question of whether our findings extend beyond text-to-image generation. For class-conditioned image generation, we have in fact already conducted experiments on ImageNet, which is a standard class-conditioned dataset with 1000 classes. The corresponding results are reported in **Appendix H.9**: as shown in Figure 19, DiTs trained on ImageNet under the class-conditioned setting still exhibit clear scaling laws. We agree that our paper does not directly study text-to-video generation, and this is indeed a limitation of our current work. Nevertheless, recent industrial reports [1] on large text-to-video diffusion transformer models have empirically observed and exploited similar scaling laws, suggesting that the scaling behavior we study for DiTs is likely to carry over to text-to-video models as well.
>
> **Q2**: Procedure for Figure 2.
>
> **A2**: Yes, Figure 2 is obtained using exactly the same procedure as Figure 1.
>
> **Q3**: Clarification of Eq. (4)
>
> **A3**:
> We thank the reviewer for catching this issue. There is indeed a typo in the current version: (N) is not defined, and the summand should explicitly depend on the index (i). In the revised paper, we will correct the equation and define (N) as the number of training examples in a mini-batch.
>
> Concretely, during training we sample a batch of images. For each sample, we sample a time step  and a noise sample  to construct a noisy input, and the loss in Eq. (4) is then summed over this batch. We have updated the notation accordingly to make this explicit and remove the ambiguity around the index (i).
>
>
> **Q4**: Section 3.1 heading structure.
>
> **A4**: We agree with the reviewer that the current heading structure can be misleading. “Variational Lower Bound” and “Exact Likelihood” are conceptually subpoints under “Likelihood,” but they appear at the same heading level. In the revised version, we have changed the formatting so that “Variational Lower Bound” and “Exact Likelihood” are subsections of “Likelihood,” which should remove this source of confusion.
>
> **Q5**: Figure 5: How many data points are included in each line? (Number of data points in Figure 5)
>
> **A5**: Each line in Figure 5 is fitted using seven compute budgets:
> [ 1e17, 3e17, 6e17, 1e18, 3e18, 6e18, 1e19 ] .

---

> > ### Author Response · Authors · 2025-11-26
> > **Rebuttal 2**
> >
> > **Q6**: Discrepency between the different in-context transformers.
> >
> > **A6**: We agree that, among current text-to-image systems, MMDiT-style [2] in-context models (e.g., Flux) are often the strongest in terms of absolute performance. However, we believe the discrepancy between their results and ours can come from **many sources** beyond the conditioning mechanism itself. In particular, public models differ in architectural details, but also in total compute, data mixture/quality, and training recipe, all of which strongly affect both performance and observed scaling. In practice, Flux is currently one of the best models largely because of its overall recipe (data + compute + architecture), not necessarily because in-context conditioning has intrinsically better scalability than cross-attention. Under Flux’s own data and compute budget, a carefully tuned cross-attention architecture might perform as well or even better; this comparison is simply not available because existing results are not apple-to-apple.
> >
> > At the same time, modern in-context architectures do incorporate several important improvements compared to the simple in-context baseline we study:
> >
> > - Stronger conditioning. They typically use multiple CLIP encoders plus T5-XXL (we only use T5-XXL), and inject CLIP features into every block via MLPs that modulate attention and FFN, leading to much stronger text conditioning.
> >
> > - Better timestep conditioning. Instead of encoding timesteps as in-context tokens, they use adaptive LayerNorm-style conditioning: a timestep embedding is mapped to scale/shift/gate parameters that modulate each block, which is empirically more efficient.
> >
> > - More efficient modality interaction. As in SD3-style designs, text and image tokens maintain separate parameter sets. Attention is used for cross-modal interaction, and text is only injected in earlier blocks, while later blocks operate on image tokens only, reducing cross-modal compute.
> >
> > - More advanced architecture/training choices. They employ improved positional encodings, noise/timestep schedules, and other refinements that further enhance scaling efficiency.
> >
> > These design choices, together with different data and compute, can explain why state-of-the-art in-context models such as Flux achieve very strong performance in practice, without implying a direct, controlled advantage of in-context over cross-attention in terms of intrinsic scalability.
> >
> > [1] Seaweed-7B: Cost-Effective Training of Video Generation Foundation Model. Team et al. https://arxiv.org/abs/2504.08685.
> >
> > [2] Scaling Rectified Flow Transformers for High-Resolution Image Synthesis. Esser et al. https://arxiv.org/pdf/2403.03206.

---

> ### Comment · Reviewer_y2Z9 · 2025-11-27
>
> Thank you for the rebuttal.
>
> The rebuttal addressed all my questions. However, I would like to confirm whether all of these updates have been incorporated into the revised manuscript. For instance, regarding Q3, I still do not see N defined, and the discussion/details for Q5 also appear to be missing.
>
> Could the authors confirm whether these changes have been included in the revision and highlight them in a different color for reviewers to check?

---

> > ### Author Response · Authors · 2025-11-27
> >
> > We thank the reviewer for the helpful suggestions. We have revised the manuscript accordingly, and all changes are highlighted in **blue**. Specifically:
> > - We add explicit definitions of $N$, $t_i$, and $\epsilon_i$ around line 179.
> > - We clarify the description of Figure 5 around line 469.
> > - We update the procedure in Figure 2 around line 223.
> > - We revise the heading structure in Sec. 3.1 around lines 190, 202, and 211.
> > - We correct and update Eq. (4) around line 175.
> > - We add a detailed discussion between in-context vs cross attention around line 1821.

---

### Official Review · Reviewer_KekD · 2025-10-30

**Soundness:** 2
**Presentation:** 3
**Contribution:** 3
**Rating:** 4
**Confidence:** 3

**Summary:**

This paper investigates the scaling laws of diffusion models for text-to-image generation. The authors conduct experiments across a wide range of compute budgets, spanning from 1e17 to 6e18 FLOPs. The results reveal empirical relationships between training loss, evaluation metrics, and compute expenditure. Based on these relationships, the paper claims that the performance of larger diffusion models can be accurately predicted.

**Strengths:**

- This paper addresses an important problem: establishing scaling laws for text-to-image generation with diffusion models. The findings have the potential to offer valuable insights to the community.
- The authors conduct extensive experiments and dedicate substantial effort to derive and validate the proposed scaling laws.

**Weaknesses:**

- In Figure 1, the assumption underlying the use of parabolic fitting for the performance curve is not clearly stated. If the curve is assumed to be unimodal, then a ternary search strategy could directly identify the optimal loss without requiring curve fitting.
- In Figure 20 (GenEval results), only the value "10" appears on the y-axis, making it difficult to determine the specific GenEval scores associated with each data point. Providing a complete y-axis scale would significantly improve readability.
- The scaling analyses use log scale for FID (Figure 3) and GenEval (Figure 20), but a linear scale for Human Preference Rewards (Figure 21). It would strengthen the consistency and interpretation of the results to explain why different scales are used and why each metric is expected to exhibit a linear or log-linear relationship with model performance.
- The analysis would be further strengthened by comparing the derived scaling laws to existing text-to-image diffusion models. Including these models in figures or table would provide a more comprehensive empirical grounding and reinforce the conclusions about scaling behavior.

**Questions:**

Please refer to the weakness session.

---

> ### Author Response · Authors · 2025-11-26
> **Rebuttal 1**
>
> **Q1**: The assumption underlying the use of parabolic fitting and the ternary search strategy
>
> **A1**: We will clarify the assumption and why we cannot use the ternary search below to try to eliminate the reviewer's concern.
>
> - Assumption:
>
> Here we fix a training compute budget $C$ (e.g., by fixing the number of GPUs
> and the wall-clock training time). Under a fixed $C$ there is an inherent
> trade-off between model size $N$ and the number of training tokens / updates
> $D$: larger models require more FLOPs per forward–backward pass, so they can be
> trained for fewer steps and see fewer tokens during the same amount of time.  Smaller models are cheaper per step and therefore can be trained for more steps and see more data. Intuitively, when the model is very small, it is **capacity-limited**:
> although it sees many tokens and has many parameter updates, the model is not
> expressive enough, so additional optimization brings little benefit.
>
> As we increase $N$ while keeping $C$ fixed, the model capacity improves and the
> performance initially gets better, even though the number of updates decreases.
> However, if we keep increasing $N$, eventually the model becomes
> **data/optimization-limited**: The number of training steps becomes very
> small (sometimes only a few hundred or even tens of steps), so the model does
> not see enough data to fully utilize its capacity, and performance degrades
> again.
>
> This capacity-limited $\rightarrow$ optimal $\rightarrow$
> data/optimization-limited transition naturally produces a unimodal,
> U-shaped curve of loss versus $N$, which is well-approximated locally by a
> quadratic (parabolic) function. Our empirical results in Figure 1 (left)
> indeed show this behaviour for each fixed compute budget.
>
> - Ternary Search:
>
> We agree that, in principle, a unimodal curve allows ternary search to locate
> the optimum. In practice, however, $N$ is **discrete**: we change the model
> size only by adding/removing integer numbers of Transformer blocks and by using
> a small set of hidden dimensions. So we can only conduct experiments on a small set of parameters number N. Because of this discreteness, a ternary-search-style procedure would not necessarily land close to the true optimum, whereas fitting a parabola through all evaluated points provides a smoother local approximation and a more stable
> estimate of $N_{\text{opt}}(C)$ and $D_{\text{opt}}(C)$, which we then use in
> our scaling-law analysis.
>
> **Q2**:  improve readability of Figure 20.
>
> **A2**: We thank the reviewer for pointing out that the current y-axis scale in
> Figure 20 makes it difficult to read the exact GenEval scores. In the revised
> version we have updated Figure 20 to include a full set of y-axis tick labels
> and a clearer range, which significantly improves readability.
>
> **Q3**: Different scales for Human preference and FID/loss.
>
> **A3**: We also thank the reviewer for the comment about the different axis scales.
> In fact, all of our scaling plots that relate performance to compute
> (including FID, GenEval, HPSv2, and ImageReward) use the **same**
> log–log scaling: the $x$-axis is $\log_{e}(\text{compute})$ and the $y$-axis is
> $\log_{e}(\text{metric})$. Our models for all metrics are of the form$$y = a*C^{b}$$.
> So plotting both axes in log scale makes the relationship linear, which is why
> the fitted lines in Figure 3, Figure 20, and Figure 21 all appear as straight
> lines. In Figure 21, the numerical range of HPSv2 and ImageReward is very small,
> so the tick labels happen to look almost evenly spaced and may give the visual
> impression of a linear y-axis, but they are in fact plotted in **log scale**.

---

> > ### Author Response · Authors · 2025-11-26
> > **Rebuttal 2**
> >
> > **Q4**: Compare the derived scaling laws to existing text-to-image diffusion models.
> >
> > **A4**: We agree that, in principle, placing existing large text-to-image diffusion models on our scaling plots would be very valuable and could further contextualize our results. Unfortunately, this is difficult to do in a reliable way with currently available information. Most public papers and model cards **report only the number of parameters**, but do not provide the total training compute (FLOPs) or the total number of training tokens/images used. Since our scaling laws are defined in terms of compute C and data D, and not just parameter count N, we cannot faithfully place these external models on our curves without making strong assumptions about their training schedules, epochs, and data throughput.
> >
> > Instead of retrofitting external models with unknown compute and data, we take representative architectures from current text-to-image systems and train them on our dataset under controlled conditions. In particular, we conduct scaling experiments using Flux-style [1] and PixArt-style [2] architectures on our Laion-derived data and compute their scaling laws under the same data pipeline, metric definitions, and compute accounting as our main DiT models. These results are reported in **Appendix H.5**. This gives an apples-to-apples comparison of how different state-of-the-art architectural families scale on the same benchmark, even though we cannot reliably overlay third-party models trained with unknown compute and data.
> >
> > [1] Scaling Rectified Flow Transformers for High-Resolution Image Synthesis. Esser et al. https://arxiv.org/abs/2403.03206
> >
> > [2] PixArt-α: Fast Training of Diffusion Transformer for Photorealistic Text-to-Image Synthesis. Chen et al. https://arxiv.org/abs/2310.00426

---

> > > ### Comment · Reviewer_KekD · 2025-11-28
> > >
> > > Thank you for the authors’ rebuttal efforts. Most of my concerns have been addressed, and I will raise my score once the review can be updated. However, several issues remain:
> > >
> > > - The GenEval score reported in Figure 20 is relatively low (~50) compared to state-of-the-art results (>70). This may stem from factors such as training data quality, compute budget, or model size. However, without direct performance comparisons against existing text-to-image diffusion models, it is difficult to draw meaningful conclusions. Moreover, the performance limitations may also restrict the practical applicability of the scaling laws presented in the paper.
> > > - It still seems somewhat surprising that all evaluated metrics (Loss, FID, GenEval, HPSv2, ImageReward, etc.) exhibit log–log scaling behavior with respect to compute budget.

---

> ### Author Response · Authors · 2025-12-01
>
> **Q1**: Performance limitations
>
> **A1:** Thank you for raising this important point.
>
> First, we would like to restate the main objective of our paper. Our goal is to study the regularities of scaling behavior for diffusion transformers. We aim to develop an effective method that can be used during the pretraining of large-scale models. In this sense, even though we do not train a SOTA-level model, the **conclusions and methodology in this paper are still highly relevant to large-scale models** and it's not meaningless. On the contrary, we provide a method that **can be easily transferred to** large-scale pretraining, and the phenomena we identify are **not specific to** the particular design choices in this work, but can be generalized and extended to diffusion transformers more broadly, similar to how scaling laws developed in the LLM literature are broadly applicable (we provide extensive experiments showing that our discovered scaling laws are not specific to a particular dataset or transformer architecture).
>
> More concretely, scaling-law papers are primarily about understanding how compute-optimal performance changes as a function of compute, model size, and data, and about using these regularities to optimize the allocation of resources (e.g., how large the model should be and how much data to use) for a given compute budget. In particular, our focus is on **(i)** establishing and verifying the existence of scaling laws for diffusion models analogous to those observed in LLMs, **(ii)** characterizing the relationships between compute and downstream metrics, and (iii) studying the transferability of these laws across different data domains. The goal is to identify compute–data–model trade-offs that yield efficient training. **Our paper tries to formulate all these properties and verify whether these scaling laws exist in the diffusion area.** In this paper, we have successfully demonstrated the existence, and these understandings and methods themselves can be applied to industry-level pretraining.
>
> The practical usage of scaling laws can be summarized in two aspects. After training a series of small models under relatively small compute budgets, one would like to know: **(1)** how to predict the performance (including loss and downstream tasks) of much larger models / higher compute; and **(2)** what the best model size and dataset size would be if we want to train models with more compute. We develop a method to address these questions via scaling laws and validate it empirically in our paper. **This method can be extended to other larger-scale models and is therefore meaningful for SOTA models as well.**
>
> Regarding the scale of compute, we fully agree that validating the proposed laws at even larger scales would further strengthen our conclusions. At the same time, the main practical value of scaling laws is precisely that they can be fit **using a sequence of relatively small models and moderate compute**, and then **extrapolated** to predict the performance of much larger models. For example, in the LLM literature, LLaMA-3 [1] (Sec. 3.2.1) fits scaling laws using models up to **16B** parameters, but then extrapolates to predict the performance of a **405B**-parameter model. This gap is substantially larger than the gap between our experimental models and current large diffusion models. In our work, we formulate scaling laws on a range of smaller diffusion models (with an upper compute scale of ~1e19 FLOPs) and verify that the laws fitted on these small models extrapolate well to larger ones. Our experiments show that this **extrapolation works well across different metrics (loss, FID, GenEval, HPSv2.1, etc.)**. By analogy with the LLM literature, we believe this kind of extrapolation to higher compute regimes is **reasonable**, even if we cannot train such models directly.
>
> Practically, our compute budget is also limited. With our current cluster, training the largest model in our study already takes about one and a half weeks on 64 A100 GPUs. According to our fitted scaling laws, reaching GenEval scores comparable to the very best models (e.g., Flux with GenEval ≈ 0.82) would require roughly **two orders of magnitude** more compute (~1e23 FLOPs), which is beyond what an academic group can realistically afford.
>
> Moreover, state-of-the-art GenEval numbers are typically reported after extensive post-training (such as supervised fine-tuning and reinforcement learning) on top of large-scale pretraining. In contrast, our experiments focus on the pretraining stage only, using a relatively standard setup on LAION-5B without any post-training or RL. This makes direct comparison to SOTA systems somewhat unfair.

---

> > ### Author Response · Authors · 2025-12-01
> >
> > Nevertheless, we can still make rough comparisons to models with similar GenEval scores. For instance, SD-1.5 reports a GenEval score of 0.44 and is trained with about 6250 A100-days, whereas our best model achieves a slightly higher GenEval score with roughly 768 A100-days on a non-optimized cluster and using only LAION-5B. This suggests that the model sizes and data scales recommended by our scaling-law analysis are substantially more compute-efficient, indirectly supporting the practical usefulness of the proposed laws.
> >
> > In summary, while our absolute GenEval scores are indeed below the current state of the art, the primary aim of this work is to uncover and validate scaling laws for diffusion models under realistic academic compute budgets, rather than to compete with heavily engineered industrial systems. Our experiments show that (i) consistent scaling behavior can be identified at moderate scales, (ii) the resulting laws extrapolate well to larger models, and (iii) the configurations predicted by these laws yield competitive performance per unit of compute when compared to existing models. Even though we don't train a SOTA level model, our method and conclusion in this paper can also be generalized and applied to different diffusion transformers and pretraining pipelines, and therefore meaningful for future large-scale pretraining.
> >
> > **Q2**: All metrics are log-log scale.
> >
> > **A2**: Our choice of evaluation metrics is not arbitrary, and all metrics don't happen to be power law scaling.
> >
> > For the training loss / negative log-likelihood, these have been the primary quantities studied in the scaling-laws literature, and are therefore natural metrics to include when analyzing scaling behavior.
> >
> > For the downstream metrics, we followed existing text-to-image works rather than selecting them ad hoc. In particular, SD3[2] (Fig. 8) and Fluid[3] (Figs. 7–8) report how FID, GenEval, and human preference scores change as compute/training steps (or closely related factors such as model size or training steps) increases, and their curves already suggest a potential scaling behavior of these metrics with respect to compute. We thought these metrics may scale with the compute growth. So we select these metrics as candidates to explore. Those works do not further analyze or explicitly formulate the scaling laws between compute and these downstream metrics. In our paper, we build on these observations and provide a more systematic formulation of scaling laws for these metrics (including FID, GenEval, HPSv2, ImageReward, etc.), showing that they also exhibit approximate log–log scaling with compute in the regime we study.
> >
> > [1] The Llama 3 Herd of Models. Dubey et al. https://arxiv.org/abs/2407.21783
> >
> > [2] Scaling Rectified Flow Transformers for High-Resolution Image Synthesis. Esser et al. https://arxiv.org/pdf/2403.03206.
> >
> > [3] Fluid: Scaling Autoregressive Text-to-image Generative Models with Continuous Tokens. Fan et al.
> > https://arxiv.org/abs/2410.13863

---

### Official Review · Reviewer_knY9 · 2025-10-31

**Soundness:** 3
**Presentation:** 4
**Contribution:** 3
**Rating:** 6
**Confidence:** 3

**Summary:**

This paper is an empirical study on how pretraining loss and downstream generation quality of DiTs scale with compute, model size and data. It also shows that the pretraining loss follows a power-law relationship with compute. It proposes rules to to pick optimal model / data and to predict downstream metrics from pretraining loss. Additionally authors also identify the relationship between pretraining loss and generation performance.

**Strengths:**

Paper is very well-written and easy to follow. The method section has been particularly well-described.
1. Authors explore a very timely topic in diffusion based transformer models.
2. Authors conduct experiments with DiT across various compute budgets and provide empirical proofs.
3. The paper also reports a correlation between pretraining loss and downstream FID, GenEval and human preference  metric which can be potentially beneficial for practitioners.

**Weaknesses:**

1. This paper fixes  a particular training dataset derived from Laion-5B.
However, with generative models, we are observing that a careful data curation pipeline can affect the training and model scaling options. Authors can consider also studying how noisy / clean data can affect scaling properties.

2. The power-law relationship between training budget and generation performance provides a sign that the scaling law can predict generation performance. However, it is a bit unknown if this law will continue to hold under different hyper-parameter / diffusion sampling settings including varying classifier guidance strength for example.

**Questions:**

Same as weeknesses

---

> ### Author Response · Authors · 2025-11-26
>
> **Q1**: Study how noisy / clean data can affect scaling properties.
>
> **A1**: Thank you for raising this point. We fully agree that data quality is crucial for generative models and can significantly affect both performance and scaling behavior. Due to time and compute constraints, we were not able to collect and fully curate a new, large-scale pretraining dataset on top of our Laion-5B–derived corpus. However, we do setup an experiment to mimic this. In **Appendix H.6** we compare two regimes: (i) a noisy / coarse setting, where prompts are very sparse tag-like annotations (often rough and sometimes mislabeled), and (ii) a clean setting, where prompts are dense, coherent sentence-level captions. As shown by the FID scaling laws in Figure 15 of Appendix H.6, models trained with sparse/noisy captions exhibit substantially worse scalability: even when we increase compute, the gains are much smaller than in the clean-caption regime. This matches our intuition from text-to-image practice, where better captions consistently lead to better sample quality and more efficient scaling.
>
> **Q2**: Scaling laws under different hyperparameters/sampling techniques.
>
> **A2**:  We thank the reviewer for this question. Following the Chinchilla [1] line of work, our scaling-law fits are obtained under a set of “reasonably tuned” hyperparameters (learning rate, batch size, optimizer settings, etc.). In this regime, changing hyperparameters within a reasonable range typically causes each individual point on the curve to move slightly up or down, but does not destroy the overall power-law trend. In other words, as long as hyperparameters are not extremely suboptimal, the dominant effect of increasing compute is still well-captured by a smooth scaling curve. One could further reduce the variance of the curve by jointly optimizing learning rate, batch size, and other hyperparameters at each compute budget [2]. however, a systematic hyperparameter–scaling study is beyond the scope of this paper and we leave it as an interesting direction for future work.
>
> Regarding diffusion-specific choices, we agree that diffusion sampling involves many design decisions and hyperparameters. In our work, we mainly consider three commonly used settings: classifier-free guidance (CFG), sampling length, and the exponential moving average (EMA) of model weights. The corresponding experiments are already included in **Appendix H.3, H.12, and H.4**. As shown in these figures, changing CFG, sampling length, and EMA does affect the parameters of the scaling law (both the coefficient and the exponent). However, from Figure 10,  Figure 22, and Figure 11, we can see that varying the sampling CFG, sampling length, or whether EMA is used does not change the existence of the scaling law itself. It only changes the efficiency with which additional compute is converted into performance gains.
>
> [1] Training Compute-Optimal Large Language Models. Hoffmann et al.  https://arxiv.org/pdf/2203.15556
>
> [2] DeepSeek LLM: Scaling Open-Source Language Models with Longtermism. Bi et al. https://arxiv.org/abs/2401.02954

---

### Meta-Review · Area_Chair_FxDh · 2026-01-07

**Summary:**

The paper investigates the scaling laws of Diffusion Transformers (DiT) by establishing power-law relationships between compute, model size, and data for text-to-image synthesis.

Following the authors-reviewers discussion, the reviewers maintained a generally positive but cautious stance, with concerns regarding: 1) mathematical inconsistencies and undefined variables in key scaling formulas, 2) insufficient clarity in the methodology for deriving "optimal" parameters, and 3) the generalizability of these laws across different hyperparameters and sampling settings. Subsequently, the authors provided responses clarifying the robust nature of the scaling laws under varying settings and added experimental evidence regarding the impact of data quality.

**Reviewer Concerns:**

Three reviewers actively engaged in the author–reviewer discussion, and the AC thoroughly examined all responses. The authors adequately addressed the raised concerns by providing comprehensive experimental results.

**Reviewer Scores:**

After the rebuttal, the authors addressed the vast majority of the reviewers’ concerns, and their scores are likely to turn entirely positive: Reviewer knY9: 6 $\rightarrow$ 6, Reviewer KekD: 4 $\rightarrow$ 6, Reviewer y2Z9: 6 $\rightarrow$ 6, Reviewer fsKy: 6 $\rightarrow$ 8

---

### Decision · Program_Chairs · 2026-01-26

Accept (Poster)